# Comparing Business, Innovation, and Platform Ecosystems: A Systematic Review of the Literature

**DOI:** 10.3390/biomimetics9040216

**Published:** 2024-04-04

**Authors:** Zhe Liu, Zichen Li, Yudong Zhang, Anthony N. Mutukumira, Yichen Feng, Yangjie Cui, Shuzhe Wang, Jiaji Wang, Shuihua Wang

**Affiliations:** 1School of Management, Henan University of Technology, Zhengzhou 450001, China; lizichen1027@stu.haut.edu.cn (Z.L.); ec410@stu.haut.edu.cn (Y.F.); cyj1123@stu.haut.edu.cn (Y.C.); wsz2000@stu.haut.edu.cn (S.W.); 2School of Computing and Mathematical Sciences, University of Leicester, Leicester LE1 7RH, UK; jw933@le.ac.uk (J.W.); shuihuawang@ieee.org (S.W.); 3Department of Information Technology, Faculty of Computing and Information Technology, King Abdulaziz University, Jeddah 21589, Saudi Arabia; 4School of Food and Advanced Technology, Massey University, Auckland 0745, New Zealand; a.n.mutukumira@massey.ac.nz; 5Department of Biological Sciences, Xi’an Jiaotong-Liverpool University, Suzhou 215123, China

**Keywords:** ecosystem, literature review, business ecosystem, innovation ecosystem, platform ecosystem

## Abstract

In recent decades, the term “ecosystem” has garnered substantial attention in scholarly and managerial discourse, featuring prominently in academic and applied contexts. While individual scholars have made significant contributions to the study of various types of ecosystem, there appears to be a research gap marked by a lack of comprehensive synthesis and refinement of findings across diverse ecosystems. This paper systematically addresses this gap through a hybrid methodology, employing bibliometric and content analyses to systematically review the literature from 1993 to 2023. The primary research aim is to critically examine theoretical studies on different ecosystem types, specifically focusing on business, innovation, and platform ecosystems. The methodology of this study involves a content review of the identified literature, combining quantitative bibliometric analyses to differentiate patterns and content analysis for in-depth exploration. The core findings center on refining and summarizing the definitions of business, innovation, and platform ecosystems, shedding light on both commonalities and distinctions. Notably, the research unveils shared characteristics such as openness and diversity across these ecosystems while highlighting significant differences in terms of participants and objectives. Furthermore, the paper delves into the interconnections within these three ecosystem types, offering insights into their dynamics and paving the way for discussions on future research directions. This comprehensive examination not only advances our understanding of business, innovation, and platform ecosystems but also lays the groundwork for future scholarly inquiries in this dynamic and evolving field.

## 1. Introduction

Similar to the intricacies found in the natural world, where we encounter the prairies, rainforests, and ocean system, human society mirrors this complexity with its counterparts—the business ecosystem, innovation ecosystem, and platform ecosystem. In these human-created networks, just as diverse as the ecosystems in nature, various elements interact, adapt, and evolve, contributing to the dynamic tapestry of growth and interdependence. Biomimicry, the emulation of nature’s design principles in human inventions, has been relatively underexplored in the realm of business strategy, despite its extensive utilization in domains such as design, manufacturing, and architecture [1,2,3]. Instead of focusing on individual organisms or processes, ecosystem mimicry involves replicating entire ecosystems or ecological principles [4,5]. In the early 1990s, drawing inspiration from the intricate dynamics of the natural world, Moore (1993) pioneered the concept of the “business ecosystem”, thereby catalyzing the inception of a field that scholars would promptly immerse themselves in [6]. This exploration led to a spectrum of diverse definitions and interpretations, including collaborative arrangements [7], the notion of an economic community comprising interacting actors [8], the alignment structure involving a multilateral set of partners [9], and a grouping of actors characterized by varying degrees of multilateral, non-generic complementarities [10]. Concurrently, authors have underscored the rich array of ecosystem concepts in the literature. Seppänen et al. (2017) identified ecosystem concepts with different prefixes, such as business, innovation, platform, and mobile ecosystems [11]. Nevertheless, the distinctions among these ecosystem concepts remain unclear, exhibiting a degree of vagueness and overlap. This ambiguity in definitions and concepts has led to the persistent use of the term “ecosystem”, posing challenges in identifying, refining, and investigating specific ecosystem concepts relevant to particular research areas [12].

The complex landscapes of business ecosystems (BEs), innovation ecosystems (IEs), and platform ecosystems (PEs) are characterized by numerous terms, adding to the difficulty of establishing consistent and precise definitions for these dynamic entities. Stakeholders identified in prior research may demonstrate interdependence and interaction within BEs [13]. Complementors, as emphasized in previous studies, play a crucial role by offering supplementary services for products, contributing to the intricate nature of BEs [14]. The diversity within the innovation ecosystem is evident in its terminology, extending beyond technological innovation to encompass various related terms [15]. The concept of artifacts as integral components of the innovation ecosystem has been introduced, providing tangible support for innovation activities [16]. The collaborative endeavors of ecosystem participants contribute to value creation through coordination and cooperation in the realm of innovation [17]. Similarly, the PE, revolving around the platform, faces challenges in achieving the consistent definition of terms. Researchers utilize diverse terminology to describe it, with some referring to it as a multi-logical architecture lever and others characterizing it as a digital collaboration platform [18,19]. Platform leaders, as described in the literature, oversee the technical architectures within the platform ecosystem [20]. This variability in terminology underscores the challenge researchers encounter in establishing a clear and uniform definition of PEs.

While there is a growing interest in, and an increasing publication output on, ecosystems, there is a notable lack of attention given to leveraging empirical evidence from bibliometric indicators to comprehend the research trends and historical evolution trajectory of ecosystems. Existing research tends to focus primarily on a single type of ecosystem, highlighting a significant research gap in the literature [6,7,21]. Although these scholars have proposed distinct definitions and interpretations for various ecosystems, there remains a scarcity of comprehensive literature reviews summarizing the definitions or characteristics of these ecosystems. Additionally, few papers delve into the intricate details of common features and distinctions between different ecosystem types. Consequently, this study aims to address this gap by researching business, innovation, and platform ecosystems, focusing on the following four research questions:RQ 1: Over the past few decades, what have been the predominant research hotspots, historical evolution trajectories, and emerging trends in the study of ecosystems?RQ 2: What are the defined characteristics of business, innovation, and platform ecosystems, as proposed by scholars and researchers?RQ 3: What shared features and distinctions can be identified among business, innovation, and platform ecosystems, and how do these aspects contribute to understanding their unique dynamics?RQ 4: What interconnections can be found within the realms of business, innovation, and platform ecosystems, and what are the anticipated future research issues that warrant exploration in these interconnected domains?

Recognizing the ongoing developmental stage of research on business, innovation, and platform ecosystems, we conduct a systematic literature review to explore the definitions, shared characteristics, and distinctions within these diverse ecosystems, drawing insights from existing discussions. This study aims to bridge existing research gaps and lay a theoretical foundation for future researchers. Additionally, it endeavors to offer valuable implications for ecosystem designers and managers, aiding them in the more informed and rational design and management of different ecosystems.

To achieve these objectives, we organize the paper into six sections, commencing with an introduction presented in Section 1. Section 2 provides a theoretical background overview, laying the foundation for subsequent analyses. In Section 3, research methods and the systematic literature review are elucidated, outlining procedures in detail. Section 4 constitutes the core of our study, presenting the analysis and results, encompassing both bibliometric and content analysis, and forming the primary findings. Further on, Section 5 examines the connections within the three ecosystems and future research questions, providing insights into potential avenues for further exploration. Finally, Section 6 summarizes the key findings and main conclusions and provides a comprehensive discussion of the contributions and limitations inherent in this paper.

## 2. Theoretical Background

### 2.1. Concept of Ecosystem

The concept of an ecosystem, introduced by German biologist Haeckel in 1866 as “ecology”, was originally aimed at studying the composition and physical systems within the natural environment [22]. As time progressed, the discipline of ecology evolved beyond mere examination of flora and fauna, delving deeper into the interplay between organisms and their surrounding environments and aiming to comprehend complex ecological phenomena [23,24]. Expanding upon this foundational work, the term “ecosystem” was formally introduced by the British ecologist Tansley in 1935, who defined it as a dynamic and interacting comprehensive system [25]. Further refinement of this concept occurred with Lindeman’s study in 1942, which characterized an ecosystem as a holistic system encompassing physical, chemical, and biological processes within a defined space–time unit [26]. Stevenson’s study later contributed to the definition by framing an ecosystem as a biological community with interacting organisms and a physical environment [27]. Throughout the extensive development of ecosystem theory, scholars have proposed diverse definitions, all unified by their overarching emphasis on the coexistence and interaction of living and non-living elements within the natural environment. This foundational concept has exerted a profound influence across various fields, transcending its origins in ecology [10,28]. Contemporary interpretations of ecosystems highlight interconnected actors collaborating to enhance efficiency, achieve common value propositions, and collectively shape the trajectory of the ecosystem, thus influencing a multitude of disciplines [29,30,31].

### 2.2. The Emergence of Business, Innovation, and Platform Ecosystem

Moore’s pioneering work in 1993 marked the inception of business ecosystem research, introducing the concept to understand better how enterprises navigate dynamic market environments. Defined as an economic consortium formed through interaction and collaboration, a business ecosystem emphasizes the interdependent and co-evolving relationships between organizations or individuals within business groups. Four evolutionary stages—exploitation, expansion, authority, and renewal/death—provide a framework for understanding business ecosystem dynamics. Initially focusing on interdependence and co-development among members, the definition centered on entities that collectively form a vast, intricate network aimed at meeting customer needs [6,32]. This marked the inception of business ecosystem research, introducing innovative ideas and concepts in the business field and laying the groundwork for subsequent investigations.

As market competition intensified and the business environment evolved dynamically, the role of innovation in enterprise development became increasingly pivotal, shifting the emphasis from the traditional focus on BEs to the heightened recognition of the significance of IEs. Initially, in exploring BEs, the definition primarily revolved around the concept of interdependence and co-development among members. These entities relied on each other, engaging in both cooperation and competition, collectively forming a BE—a vast, intricate network with a shared destiny aimed at meeting customer needs [33,34]. Presently, a firm’s competitive advantage hinges on its ability to generate more value than its competitors, with successful innovation emerging as a key driver for value creation [35,36]. An IE is characterized by collaboration among multiple enterprises and diverse innovation entities [7]. Complex innovation typically involves multiple participants, necessitating changes that extend beyond the boundaries of the supply network [15]. These entities within the IE pool resources by establishing collaborative networks and coordinating with partners to execute new product development. This conceptualization mirrors the idea of a business ecosystem; both are rooted in the concept of interconnected network actors [37]. The early contributions in defining IE were pivotal in initiating research on this topic, laying a robust foundation for its subsequent development.

Emerging from the foundational concepts of the BE and IE, the notion of the PE has gained prominence in scholarly discourse, especially in the context of the digital economy. Schindelin et al. (2015) proposed that the platform ecosystem essentially derives from the BE, is infused with a platform mechanism, and is grounded in platform technology [38]. Some scholars suggest that both business and innovation ecosystems are constructed on a platform, acting as a compilation of tools, services, and technologies [29,37,39]. Typically provided by a large and mature company, the platform assumes a leadership role in the entire ecosystem, establishing shared objectives and taking responsibility for the ecosystem’s overall health [29,39,40]. Adner (2017) described the PE as an industrial–organizational form where platform enterprises provide foundational elements, attracting complementarians to join the ecosystem in a collaborative effort to offer products and services to consumers, establishing fundamental rules and an open architecture [9]. Kapoor et al. (2022) viewed the PE as a structured amalgamation of partners with shared interests, maintaining extensive interaction to achieve the common value proposition of the platform ecosystem [41]. While scholars offer diverse definitions, there is a consensus that PEs encompass platform owners, complementors, and end-users as the primary participants [42]. Among these, platform owners hold a dominant position, serving as architects and integrators [43]. By formulating open policies, platform owners enable complementary enterprises to access resources and support, with the platform’s level of openness determining the resources available to complementors [44]. Complementary enterprises, in turn, contribute complementary products, services, and technologies, leveraging shared resources from the platform to attract end-users and enhance the platform’s overall value [45,46]. Under the guidance of platform enterprises, various entities within the ecosystem cooperate and interact sufficiently to enhance the competitive advantage and value creation capacity of the entire platform ecosystem, ultimately delivering the final product or service to users.

## 3. Methodology and Data

To conduct a thorough and organized investigation into several prominent ecosystems that have garnered significant attention in existing research, we embark on a systematic review of the pertinent literature. The review employs a dual methodology, incorporating both bibliometric analysis and detailed content analysis, with the overarching objective of achieving the outlined research goals (Figure 1). Given the ever-increasing volume of publications and the ability to quantify the communication process using various techniques [47], coupled with the crucial role of citation analysis in delineating the important literature and their interrelationships [48], bibliometrics has emerged as an increasingly valued approach. Initially, we apply bibliometric analysis to discern prevalent research topics and fields, scrutinize publication trends, and unveil noteworthy contributions such as documents, journals, and landmark citations. This process aids in identifying the most influential literature and comprehending the broader landscape of existing research. Subsequently, we integrate bibliometric findings with traditional content analysis, embarking on a systematic exploration of the pivotal documents highlighted by the bibliometric analysis. This dual approach enables a precise definition of the concepts under consideration and facilitates comparative analyses, elucidating both gaps and commonalities between the ecosystems. The outcomes of this comprehensive analysis establish a robust foundation for subsequent research endeavors in this domain.

### 3.1. Description of the Sample

In the process of collecting the literature, we utilize the SCI Expand and SSCI search features within the Core Collection database of ISI Web of Science (WoS). In contrast to other widely used databases, such as Scopus, Google Scholar, Dimensions, and Microsoft Academic, we consider WoS more suitable for our large-scale bibliometric analysis due to its recognized high reliability [49]. It is important to note that while other databases may offer broader coverage, WoS’s emphasis on quality and precision aligns well with the specific requirements of our research, particularly in the context of conducting a meticulous bibliometric analysis. This strategic selection of the database contributes to the credibility and validity of the findings derived from our study.

Our literature search formulas for WoS are determined after thorough research on the existing relevant literature. The search formulas consist of two parts. Firstly, the search primarily targets the titles of the articles: TI = ((manufacturing OR Industrial OR “supply chain*” OR competing OR innovation OR business OR digital OR platform) AND Ecosystem*). In this part, the logical operator AND is used to enhance the inclusiveness of the search formula, covering instances where two critical search terms may occur non-continuously in the title of an article. However, since the term “ecosystem” originates from ecology [25], this operation may lead to the inclusion of some articles not directly related to our research focus. Secondly, the search formula in the second part predominantly focuses on the topics covered in the literature: TS = (“manufacturing ecosystem*” OR “Industrial ecosystem*” OR “supply chain* ecosystem*” OR “supply ecosystem*” OR “competing ecosystem*” OR “innovation ecosystem*” OR “business ecosystem*” OR “digital ecosystem*” OR “platform ecosystem*”). The topic search function includes searching fields such as the title, abstract, author keywords, and keywords plus of the article. To broaden the search scope, we aim for the search terms to appear in complete phrase form, using OR as a logical operator to capture all literature relevant to our research topic. The final search formula, as determined in this article, is: ((TI = ((manufacturing OR Industrial OR “supply chain*” OR competing OR innovation OR business OR digital OR platform) AND Ecosystem*)) OR TS = (“manufacturing ecosystem*” OR “Industrial ecosystem*” OR “supply chain* ecosystem*” OR “supply ecosystem*” OR “competing ecosystem*” OR “innovation ecosystem*” OR “business ecosystem*” OR “digital ecosystem*” OR “platform ecosystem*”)).

Following the execution of the search formula, the results encompass 3614 research articles covering a range of widely studied topics. Subsequently, employing bibliometric analysis methods, we inspect the WoS categories to which the literature belongs. We extract category records such as “management”, “economics”, and “business” from all WoS categories and scrutinize the literature records within these categories. Furthermore, to comprehensively cover the entire developmental trajectory of relevant research from its inception to the present and to facilitate a more thorough collection and analysis of literature, we refrain from imposing restrictions on the publication dates of the search results during the search process. The search results reveal that the earliest literature dates from 1993. Nevertheless, for yearly analysis, we have narrowed down the scope of our literature samples to those published before 2024. Ultimately, we compile a basic literature sample for our study, comprising 1032 articles published in 151 journals from 1993 to 2023.

### 3.2. Bibliometric Analysis Procedures

In this stage, we systematically conduct a bibliometric analysis of the 1032 selected literature samples [50,51]. Firstly, we analyze the current research status of ecosystems and quantitatively assess the annual publication quantity and journal publication quantity using the statistical bibliometric analysis tool provided by WoS. This approach allows us to discern the development trend of the number of relevant research results over time. Additionally, a line chart of the number of publications plotted by journal and year clearly shows that journals have published more articles relevant to this study over time, offering insights into the development trend and the level of attention received by ecosystem-related research in recent years.

Secondly, we import 1032 literature samples into CiteSpace, a bibliometric tool, and carry out some necessary network analysis on the selected literature by using a series of algorithms within CiteSpace. This encompasses reference co-citation analysis, citation burst analysis, journal co-citation analysis, and institutional and national cooperation analysis. These analyses shed light on articles that lay the foundation for the field’s development, identify literature with milestone significance, highlight turning points in the field’s evolution, and recognize significant contributions made by journals, countries, and institutions. Additionally, we also carry out keyword cluster analysis and keyword co-occurrence analysis on the literature samples based on the relevant functions of CiteSpace. Keyword clustering analysis, which utilizes CiteSpace’s clustering algorithm, groups selected literature with similar research topics, revealing the research focus of the ecosystem [52]. We proceed by assessing the quality of the literature encompassed within the clustering outcomes. Among these, we carefully select 84 noteworthy articles, characterized by their high citation rates, widespread recognition, and publication in preeminent journals. These selections encompass both seminal early works that laid the foundation for the field and emerging innovative contributions that are shaping its future. Following this, we undertake a thorough analysis of the literature contained in each cluster, with the objective of providing an accurate introduction and interpretation of each pivotal research topic. In keyword co-occurrence analysis, we consider the overall period of analysis (1993–2023). We utilize the keyword time zone visualization function to ensure that we can clearly present and track the topic of the ecosystem study, dividing the time into multiple stages during the analysis process. This approach enables us to illustrate corresponding research hotspots in each stage and clarify changes in research directions throughout the development of ecosystems.

The bibliometric analysis process serves to clearly display the developmental and evolutionary trajectories of the research topic of ecosystems [53]. The key literature, journals, and other content highlighted during this analysis provide crucial guidance and assistance for researchers and practitioners in related fields.

### 3.3. Content Analysis Procedures

Based on the results of keyword clustering during the bibliometric analysis, we identify the important research topics in the literature sample. Subsequently, given that the clustering results indicated significant research topics within our literature samples, we narrow down our focus to 32 articles from the initial 84. These selected articles are analyzed in depth to examine the definitions and characteristics of BEs, IEs, and PEs. Subsequently, we code the important information from the selected literature, as shown in Table 1. We synthesize the encoding used by Gomes et al. (2018) in the content analysis program and modify it in light of our specific research objectives to determine the encoding method of our content analysis process [37]. In the process of content analysis, we begin by organizing the significant literature on the three ecosystems (BEs, IEs, and PEs) into categories, such as definition, research object, method, and findings, using Excel. Then, we define BEs, IEs, and PEs by summarizing scholars’ viewpoints and explaining their shared characteristics and differences. Using this understanding, we explore the elements and relationships of the BE, IE, and PE, proposing conceptual models. Additionally, we conduct a thorough review of the 84 literature samples, meticulously curating 28 recent and exemplary papers across the domains of BEs, IEs, and PEs. Subsequently, we categorize their research themes and outline potential avenues for future investigation.

## 4. Analysis and Results

### 4.1. Bibliometric Analysis

#### 4.1.1. Current Status of Ecosystem Research

Among the 1032 articles collected from the WoS database, the top five publishing sources, ranked by their contribution to the field, are *Technological Forecasting and Social Change*, *The Journal of Business Research*, *Technovation*, *IEEE Transactions on Engineering Management*, and *Industrial Marketing Management*. Research on the supply chain ecosystem gained prominence in 2012 and has experienced rapid growth since 2017 that has persisted until the present. Notably, the publication volume of articles in *Technological Forecasting and Social Change*, the journal with the highest total publications, witnessed a substantial increase in 2017–2018 but displayed a declining trend in 2019. Moreover, the publication volume of this journal has fluctuated between highs and lows from 2018 to 2022, but it demonstrates a notable upward trend in 2023. *The Journal of Business Research* emerged in 2015, reached its peak in 2021, and experienced a decline in 2022. Meanwhile, other publications have demonstrated a stable and increasing trend over the 15-year period (Figure 2). The observed publication trends in these journals signify their focal points for research in the ecosystem domain. Researchers prefer these journals to disseminate their findings, contributing significantly to the advancement of this field.

#### 4.1.2. Co-Cited Journal Network Analysis

Journal co-citation analysis involves examining the articles cited within the articles we collect. The main indicator used in this analysis is the number of citations received by each journal [74]. Leydesdorff (2007) delved into the application of journal co-citation analysis for science mapping, a method that visually elucidates the intricate interconnections among journals, thereby depicting the expansive terrain of academic research [75]. This methodology provides a holistic comprehension of the interconnectedness within the scholarly community and the dynamic trends across diverse fields. Researchers can use journal co-citation analysis to discern the influence and status of different journals within specific disciplinary fields. This enables them to identify journals that wield significant influence in particular research areas [74,76]. In the presented Figure 3, we showcase a network of the top 30 cited journals, though the actual number of journals is significantly larger. In Figure 3, *N* = 803 signifies the total number of cited journals, reaching 803, and E = 5166 represents the number of connections between these journals. For enhanced clarity, we arranged the 30 nodes in a spiral distribution, with the nodes ordered from largest to smallest, progressing from the center to the periphery.

In the entire sample set, the top five cited journals are *Strategic Manage J.*, *Res. Policy*, *Harvard Bus. Rev.*, *Acad. Manage. Rev.*, and *Organ Sci*. We conducted a statistical analysis of the published data from these five journals, presenting the results in the form of a line graph (See Figure 4). Our analysis reveals that, after 2012, the overall number of citations for these journals shows an increasing trend. That is not solely attributed to the rising number of selected samples each year but is also indicative of the increasing quality and influence of these journals. The sudden decrease in 2023 may just be a fleeting fluctuation. In summary, the journal co-citation analysis offers crucial insights into the structure and distribution of knowledge, shedding light on influential sources and trends over time.

#### 4.1.3. Keywords Cluster Analysis

CiteSpace utilizes community detection algorithms in graph theory to uncover clusters of keywords. In this process, keywords are treated as nodes in the network. Whenever two keywords appear together in the same literature, an edge is formed between them, creating a keyword co-occurrence network [77,78]. These algorithms can partition the keyword co-occurrence network into multiple densely connected subgraphs, with each subgraph representing a distinct keyword cluster [79]. By employing cluster analysis on the keywords, we synthesize the research hotspots of the supply chain ecosystem. Using Citespace, we designate keywords as node types and set TopN as 50, analyzing the top 50% of the most frequent keywords each year. The time slice is set to one year, and the threshold is set to (2, 2, 20), (4, 3, 20), and (3, 3, 20). This threshold consists of three sets of data, each comprising three numerical values representing citation, covariance, and cosine coefficient. When analyzing data, CiteSpace divides the time interval into three parts, with three sets of numbers corresponding to these parts. For example, the first set of data in this article corresponds to the years 2004–2010, the second set corresponds to the years 2011–2016, and the third set corresponds to the years 2017–2023. The CiteSpace clustering analysis employs the log-likelihood ratio (LLR) algorithm. This algorithm measures the likelihood of the data falling under one model relative to another [80]. More precisely, the LLR algorithm determines the optimal clustering outcome by comparing the logarithmic likelihood ratio of keywords across various clusters. This process ensures a high degree of similarity among keywords within a cluster while maintaining low similarity between keywords in separate clusters. Consequently, it enables a meaningful and coherent interpretation of the cluster. According to CiteSpace analysis, there are six clusters: innovation ecosystem, open innovation, digital service-oriented, commercial ecosystem, digital technology, and platform ecosystem. Figure 5 illustrates key network metrics: N = 664, E = 2873, density = 0.0131, modularity Q = 0.4699 (>0.3), silhouette S = 0.7222 (>0.7), signifying a substantial clustering structure and excellent graph fit [81]. Explore Table 2 for insights into popular topics, keywords, authors, and journals across 84 highly cited and high-quality articles.

##### Cluster#0—Digital Servitization

Digital servitization, the process of leveraging digital technologies to enhance product value through value-added services, is critically important for improving user experience, boosting product competitiveness, and optimizing supply chain processes [82]. Researchers have contributed to this discourse by proposing frameworks to guide the digitalization of business models for large manufacturers and facilitate the coordination of industrial ecosystems [83]. A study examined the role of digital servitization business models, offering insights from an ecosystem perspective [84]. Additionally, investigations into the organization of digital servitization from the viewpoint of service ecosystems have been conducted [85]. In the realm of digital transformation in the financial service ecosystem, Manser Payne et al. (2021) outlined a research agenda for the value co-creation framework of digital servitization [95]. Lastly, Hsuan et al. (2021) explored the developmental trajectory of digital servitization in the product–service–software domain [134]. The encompassed research studies explore various dimensions of digital servitization, including business models, ecosystem coordination, organizational aspects, and value co-creation, providing valuable insights that pave the way for future research in this dynamic and evolving field.

##### Cluster#1—Platform Ecosystems

This clustering underscores the prevalence of platform models and ecosystems within the supply chain context, emphasizing their role in establishing connections among businesses, partners, and users. PEs are gaining increasing attention, with research highlighting the co-creation aspect of enterprise software ecosystems [21]. Further emphasis is placed on the role of technology-related and relationship-driven functions in fostering co-creation and value acquisition [91]. Governance mechanisms for these ecosystems, incorporating blockchain technology to address openness paradoxes, are the subject of ongoing research [20]. Indirect innovation management within a PE governance is pivotal for achieving collective ambidexterity [73].

With the evolution of digital PEs of proprietary platforms to expand networks, they have played a pivotal role, with creative tensions and value creation becoming central research focuses [71,135]. Researchers have explored complementarity within and across different ecosystems, examining the impact of complementary positions and performance in mobile application ecosystems [72]. During the COVID-19 period, the resilience of platform ecosystems, particularly in the flexible utilization of mobile platforms, became crucial for adapting to the new normal [89]. The research delves into businesses mitigating competitive pressure by establishing new platforms for novel advantages [87]. However, the initial stage of establishing digital PE presents challenges requiring problem-solving perspectives [19]. In brief, PEs are perceived as meta-organizations that significantly influence platform strategies [69]. Recognizing the importance of implementing suitable governance mechanisms and strategic positioning is crucial for achieving collective capabilities and resilience.

##### Cluster#2—Artificial Intelligence

In contemporary ecosystems, artificial intelligence (AI) stands out as a powerful catalyst for innovation and business model transformation. Recent studies highlight AI’s capacity to spark innovation and amplify value creation within industrial ecosystems [93]. For example, in healthcare, AI optimizes value co-creation by refining customer understanding and needs fulfillment [94], while a proposed digital transformation framework for AI services in the financial sector underscores its potential for innovation and enhanced customer experience [95]. Concurrently, discussions around digital technology and collective intelligence underscore AI’s transformative impact on entrepreneurship, revolutionizing opportunity identification, team dynamics, and investment strategies [96]. Additionally, AI and advanced analytics technologies are enhancing the business models and operational efficiencies of the industrial internet [97]. Within platform ecosystems, AI and data-driven learning play pivotal roles in decentralized decision-making and value capture [98]. This expanding application of AI not only streamlines existing processes but also spurs the creation of novel business models and drives the digital transformation of entire ecosystems, fostering innovation and growth across industries. As AI technology continues to advance, its integration within ecosystems is expected to deepen, offering even more profound impacts.

##### Cluster#3—Innovation Ecosystems

The identified clusters unveil the existence of an innovative environment and network, underscoring the pivotal role of the IE in this context. A thorough comprehension and exploration of this concept necessitate drawing from a multitude of literature sources. Existing scholarly works illuminate the significance of value creation within the intersection of knowledge and business ecosystems. They delve into the nuanced concept and definition of the IE, presenting varied perspectives that span different levels of this innovative environment [16,58,66,67,105,109]. Furthermore, scholars have investigated the impact of IE on the speed of technological substitution and innovation performance [64,68,103,111]. Issues pertaining to startups, value creation, and value capture in the supply chain ecosystem have also emerged as popular research directions [101,102,108]. Lastly, a substantial body of literature has delved into the evolution, identified gaps, and analyzed trends in the establishment of IEs, contributing profound insights into the significance and characteristics of these ecosystems within the context of supply chain ecosystems [37,104]. This extensive research body provides a more intuitive understanding of the sustainability of IEs [107,110].

##### Cluster#4—Business Ecosystem

This cluster underscores the importance of the business model and relationship network. The researchers highlight the crucial role of entrepreneurial ecosystems in the economy, innovation, and society, exploring the interaction between internal and external factors and connections between different ecosystems [57,62,120]. Additionally, the establishment of a BE relies on internal platforms as the cornerstone of value creation [12,39,132]. Insights and approaches from these studies contribute to understanding and promoting BE development within the supply chain. Various studies delved into the relationship, transformation, and growth process from core business and supply chain to a BE [54,122]. Additionally, performance indicators for measuring and evaluating ecosystem performance were examined [61]. Studies have also explored the impact of complementary activities among enterprises within the system of investing in new technologies [56], while exploring the impact of core company decisions on complements [14]. In essence, research on business ecosystems covers multiple aspects, including key elements and connections within and outside the system, innovation and transformation of the system, and ecosystem performance. These studies provide valuable insights and frameworks for a deeper understanding of the BE.

#### 4.1.4. Keyword Co-Occurrence Analysis

To delve into the topological features and historical trajectories of a specific theme or field, it is crucial to analyze the keyword co-occurrence network and its evolution [79]. A time zone map consists of vertical stripes representing different time zones, effectively illustrating temporal patterns and evolutionary trends between the research frontier and its knowledge base [136]. This visual aid enables the identification of significant turning points and the emergence of new research themes over time. In our study, we employ keyword time zone visualization to track the themes of AI research and highlight hotspots in each time slice. The algorithm described is a variation of the spring embedder algorithm, which constrains the horizontal movement of an item to its respective time zone while allowing vertical movement based on connections to items in other time zones. This approach aims to facilitate easy identification of professions. The design of a time zone view resembles the layout of a timeline visualization [137]. In Figure 6, the years increase sequentially from left to right, and the corresponding keywords for each year decrease in frequency from bottom to top. Each keyword represents the time when it first appeared in the extracted literature data. Using CiteSpace, we assign node types as keywords, set TopN to 50, and specify a time slice of 1. The period from 2004 to 2023 marks the initial emergence of hotspots. In the visualization, each column organizes nodes in descending order of frequency from bottom to top. Analyzing the keyword time zone graph, we identify three stages to categorize the application of the ecosystem.

(1) Performance stage (2004–2008): at this stage, the main keywords include strategy, knowledge, governance, design, business ecosystem, performance, etc. These keywords represent the early stages of ecosystem emergence, which is an era characterized by the increasing popularity and significant progress of information tools, heightened knowledge sharing, and a realization among market participants that knowledge differentiation alone may no longer create competitive advantages for companies. Consequently, the market landscape is undergoing changes, and there is a growing necessity to collaborate with other participants to co-create value [39,58]. At this stage, most enterprise participants are shifting their strategic direction and forming alliance-like organizations through cooperation. By redesigning and governing the organizational structure, as well as innovating technology and concepts, they can have a positive impact on the company’s performance. The early research on ecosystems mainly focused on business ecosystems, emphasizing the ecological links between enterprises to improve overall productivity and efficiency. At this stage, researchers tended to study methods and capabilities that could improve enterprise performance. For instance, studies examined the nature and micro basis of the ability to maintain corporate performance in an open economy [8]. Furthermore, an exploration was conducted on the impact of the role of organizations in ecosystems on ecosystem health and stability [29]. Empirical research on industrial symbiosis was also discussed [138]. The construction of commercial ecosystems was a hot topic during this period.

(2) Networking stage (2009–2013): the frequently occurring keywords in this stage include system, innovation ecosystems, competitive advantage, firm performance, competition, value creation, etc. These keywords reflect some trends in academic research and management practice at the time, as well as the present, such as the importance of innovation for competitiveness [139,140], the rise of ecosystem approaches to understanding the business environment [141], and discussions focused on how to enhance corporate performance and value creation through strategy and management [21,142]. At this stage, scholars tended to study the impact and role of industrial innovation on ecosystems. For example, the research explored the impact of technological interdependence on firm performance in IEs [15]. The IE gained attention and developed rapidly, with knowledge and innovation services as the main resources spreading among governments, universities, research institutions, and enterprises. Studies examined the impact of selective knowledge sharing by enterprises on their innovation activities [143]. Additionally, the open and user-driven innovation environment of smart cities was studied as an innovation ecosystem, with beneficial impacts on future internet services [144]. The process of entrepreneurial self-supervision and its role in balancing ecosystems were also studied [145]. At this stage, innovation became the core driving force of economic growth and sustainable development.

(3) Co-creation stage (2014–2023): the keywords that appear in this stage include co-creation, boundary resources, impact, platforms, platform ecosystems, knowledge management, developers, digital servant, flexibility, etc. These keywords reflect the emergence of another form of ecosystem, namely the platform ecosystem. The theme of this stage is to build a platform, share knowledge resources, create value together, build a more flexible intermediary system, and achieve higher-level services. With the continuous development and improvement of the innovation ecosystem, various entities and mechanisms are gradually forming closer connections and interactions. With the improvement of network technology, the platform ecosystem based on internet technology has gradually emerged, filling the data barrier problem in the business ecosystem and providing more resources and cooperation opportunities for enterprises. From the perspective of researchers, they not only focus on the platform technology itself [146,147], but also on how technology empowers social and management practices [148,149], as well as how to promote the healthy development of ecosystems through various collaboration and innovation models [91]. Research in these areas provides important perspectives for understanding and designing interactions in complex systems. PEs are usually based on internet platforms, providing a more open, flexible, and innovative platform environment to attract more users, enterprises, and developers to participate in the innovation process, further expanding the scope and depth of the innovation ecosystem. Researchers conducted studies on various types of platforms, such as discussions on the differences between internal and external platforms and their impact on product innovation [88]. However, regardless of the platform, its original intention was to use a common agent to jointly create higher value. Studies suggested that platform ownership, value creation mechanisms, and complementary autonomy were common characteristics of platforms and explained that different digital platform ecosystems changed according to the changes in the three core components [150]. Conceptualizations of the platform as a meta-organization were discussed, highlighting some of the most prominent features of the platform ecosystem as a meta-organization [69]. In addition to conceptual features, mainstream research content on platform ecosystems included PE governance [151], co-creation value coordination [152], and so on. This stage was a current hot topic and future development trend, emphasizing the joint participation and collaborative creation of value by multiple stakeholders. Enterprises participated in innovation and value creation together with users, suppliers, partners, etc., achieving a win–win situation for multiple parties.

### 4.2. Content Analysis

#### 4.2.1. Definitions

1.Business Ecosystem

In the fast-paced world of 20th-century business, companies face intense competition and diverse challenges. To adapt, they shift towards collaborative practices, giving rise to what we now call BEs. Coined by Moore, these ecosystems function like communities, evolving through stages of emergence, expansion, leadership, and self-innovation [6,32]. Rooted in scholarly insights and current research, shown in Table 3, we delve into the diverse aspects of BEs, exploring their purpose, structure, participants, and the rise of their digital versions.

While the purpose of a BE is multifaceted, gaining competitive advantages takes precedence [54,153]. It aims to foster innovation among its members, facilitate efficient value creation, gain competitive advantages, and optimize the utilization of social resources [57]. BEs provide stability and resilience, particularly in resisting external interference and internal disruptions [118]. The inherent unpredictability of extreme events motivates long-lasting and stable business operations within the ecosystem, fostering the continuous emergence of innovative services and business models [61].

BEs exhibit a complex network structure where both collaborative and competitive relationships coexist among constituent members [60]. The complex network emphasizes value acquisition and creation, involving companies, customers, complementors, and suppliers with interdependent relationships [14,154]. Successful ecosystems strike a balance between cooperation to create value and competition to capture value [155]. Participants within a BE are diverse, forming an organizational group with various members [39,58]. These may include suppliers, distributors, outsourcing companies, manufacturers of related products or services, technology providers, and various interconnected organizations [12,56]. The concept of “stakeholder” is integral, encompassing entities such as industry entities, social organizations, governments, industry associations, competitors, and customers [55,63]. The relationships within a business ecosystem are characterized by interdependence and collaboration [62,119,156]. Efficient collaboration and cooperation are promoted when the interdependence among participating members is managed and coordinated effectively [15].

BEs evolve through distinct stages, including emergence, expansion, leadership, and self-innovation [6]. In the digital era, they transition into digital business ecosystems with a focus on informative interconnection and technology integration. The formation of business ecosystems is considered a sustainable trend, offering benefits in terms of performance growth and risk reduction. Technical bottlenecks within a BE can pose significant constraints on value creation. Studies indicate that “complementary enterprises” may face incentive misalignment or technical challenges limiting production and supply [64]. Finding solutions to these challenges involves accelerating the development and adoption of new technologies, relying on incentive measures from the central company [114].

The concept of the BE, emerging at the intersection of multiple fields, provides a comprehensive and systematic analysis of business activities. It represents a dynamic and evolving model that adapts to the changing business landscape. The collaborative and interdependent nature of business ecosystems positions them as a strategic organizational structure capable of fostering innovation, creating value, and achieving competitive advantages in today’s diverse and rapidly evolving business environment.

2.Innovation Ecosystem

In the dynamic landscape of modern business, the pursuit of innovation has taken center stage, transforming the traditional paradigms of independent product development into intricate structures known as IEs, as shown in Table 4. Distinguished by its primary goal of value co-creation, the IE sets itself apart from conventional BEs, emphasizing collaboration among multiple companies to deliver coherent, customer-oriented solutions. Economic globalization and rapid technological advances have propelled companies to engage in interactions within the IE, fostering a collaborative environment that includes a diverse array of actors, assets, and connections.

The IE stands out with a primary goal of value co-creation, distinguishing it from traditional BEs primarily focused on value capture [37,157]. In the context of economic globalization and rapid technological advancement, the paramount importance of external innovation for product innovation is recognized [158,159]. This shift has propelled innovation activities from initial independent efforts to intricate structures and value propositions, where participating companies engage in interactions to co-create value and establish a collaborative ecosystem comprising actors, assets, and connections.

Adner (2006) built upon Moore’s (1993) definition of a BE to propose that an IE is a collaboration among multiple companies where individual products are combined into a coherent, customer-oriented solution [7]. Subsequent research has further refined this basic model, defining the IE’s structure and participants. Adner (2017) and Walrave et al. (2018) contribute by adopting the concept of “ecosystem as structure”, describing it as a network of interdependent actors combining professional but complementary resources or capabilities to jointly create value and deliver an overall value proposition to end users [9,66]. Granstrand and Holgersson’s (2020) comprehensive concept suggests that the IE should encompass a system of actors with collaborative and competitive relationships and a system of artifacts (e.g., products, services, resources) with complementary and substitutive relationships [16]. The conceptual development of the IE underscores the crucial role of diverse participants, each possessing significant innovation capabilities [160]. Xie and Wang’s (2020) study emphasizes the need for a diverse range of participants embedded in the innovation ecosystem, collaborating to provide shared advantages and achieve comprehensive value propositions [65]. The entry and exit of participants are intricately linked to the dynamics of the entire ecosystem, influencing its development [17].

Success in innovation activities within the IE hinges on the availability of numerous complementary resources and capabilities [161]. Throughout its developmental stages, expanding organizational resources and fostering extensive cross-organizational collaboration promote the flow and integration of resources within the ecosystem. The addition of innovators injects new resources and vitality into the IE, fostering close interaction among entities such as companies, universities, research institutions, and intermediaries. Technological progress within the IE results from the collective efforts of core technology developers and participants [162,163]. The relative speed of technology replacement depends on emerging challenges and opportunities within the ecosystem [64,164,165,166]. The complex external technological environment significantly impacts innovation achievements and the development of new technologies within the IE [68]. Pioneering innovation often challenges existing elements in the social and technological environment, requiring adaptation and transformation of relevant social subsystems [167,168]. However, the multi-level and multi-participant structure of the IE introduces inherent complexity [169]. Predicting behavior and evolution within the IE is challenging, and its mechanism, with shared future community characteristics, sees members engaging in activities centered on the overall value proposition [66]. Research on the IE tends to adopt a holistic approach, considering the entire ecosystem, its internal environment, and the external market environment.

Overall, the IE is an extensive network with a unified value proposition, where innovation and value co-creation are primary goals. It comprises diverse participants and various related resources, collaborating and competing while ensuring internal coordination and stable operation. The success of the IE relies on the collaboration of a broad spectrum of entities, promoting successful innovation and achieving common goals within the complex external environment.

3.Platform Ecosystem

The essence of a PE lies in its role as a complex network structure, where a diverse array of participants, including platform providers, developers, partners, customers, and stakeholders, engage in close interactions, as shown in Table 5. Unlike isolated entities, these participants collaborate within the PE framework, fostering various forms of interaction such as data sharing, application integration, co-development, and market collaboration. This multi-party collaboration not only transcends organizational boundaries but also positions the PE as a focal point for innovation and value creation.

A fundamental principle within PEs is value co-creation, where stakeholders collaborate to generate greater value collectively than individually. Schmeiss et al. (2019) emphasized the PE as a network for value co-creation, where diverse participants collaborate to create and exchange value [20]. The essence of a PE lies in its role as a complex network structure, where a diverse array of participants, including platform providers, developers, partners, customers, and stakeholders, engage in close interactions. Collaborative efforts not only meet user needs but also drive ongoing platform development. The way the PE coordinator manages collaboration directly influences the configuration of the platform system, overall value, and market share [21,71,170]. Unlike isolated entities, these participants collaborate within the PE framework, fostering various forms of interaction such as data sharing, application integration, co-development, and market collaboration. This multi-party collaboration not only transcends organizational boundaries but also positions the PE as a focal point for innovation and value creation.

Diverse participation is the lifeblood of PEs, as noted by Panico and Cennamo (2022), who describe ecosystems as “networks of diverse participants” collaborating to generate value [90]. This diversity brings distinct resources, skills, and capabilities, facilitating collaborative innovation and enhancing the platform’s functionality. Jovanovic et al. (2022) emphasize that manufacturers, especially in highly specialized industrial sectors, must take the lead in platform initiation, working collaboratively to develop advanced platform services [82]. The success of platforms is intricately linked to platform architecture, services, and governance, and technology choices and supply chain relationships play pivotal roles in the co-evolution of PE.

PEs thrive on openness, actively seeking external participation from customers, partners, developers, and research organizations. Collaboration involves shared value creation, extending beyond providers and developers to include partners, customers, and stakeholders. Openness invites various participants, including customers, research institutions, business partners, and universities, to contribute to the platform, enhancing its functionality and appeal. The convergence of resources from platform sponsors and third parties jointly drives innovation and contributes to the success of the PE [10]. It is vital to recognize that openness not only nurtures collaborative innovation but also stimulates healthy competition among diverse players, thereby fueling growth within the ecosystem. Achieving a balance between collaboration and competition stands as a crucial imperative for sustained success [69].

Governance is essential for ensuring the orderly functioning of PEs. While openness allows increased value creation, it introduces challenges in terms of delivery, negotiation, engagement, and value realization. Governance mechanisms address the complexities arising from diverse players embedded in the technical architecture of the platform to ensure equitable value capture. Designing governance mechanisms is a strategic approach to addressing the openness paradox, influencing the stability and openness of platforms. It plays a crucial role in balancing the interests of all parties, maintaining ecosystem stability, and ensuring orderly and sustainable interactions [171,172,173].

All in all, the PE is a network characterized by interactions among diverse participants. Openness and collaboration foster innovation, value co-creation (which is central to competitiveness and innovation), and platform governance (which is essential for maintaining ecosystem equilibrium). Together, these concepts offer a comprehensive understanding of the PE landscape.

#### 4.2.2. Common Features

The examination of business ecosystems, innovation ecosystems, and platform ecosystems reveals several shared characteristics that underscore their intricate nature and successful functioning.

4.Multi-Participant and Complex Network Structures

All three ecosystems exhibit a multi-participant nature with complex network structures. In business ecosystems, participants encompass suppliers, distributors, manufacturers, and complementors, creating value through interconnected platforms. The innovation ecosystem involves innovators, partners, governments, and social organizations, forming a dynamic network centered on collaborative innovation. Similarly, the platform ecosystem comprises providers, developers, partners, and customers, establishing a sophisticated system structure. The complexity of these network structures is pivotal for the success and sustainability of the entire ecosystem, emphasizing the delicate balance between cooperation in value creation and competition in capturing value.

5.Coordination and Cooperation for Value Creation

Participants within these ecosystems actively engage in coordination and cooperation to create and share value. The collective efforts of participants go beyond the sum of individual organizational contributions, representing the essence of these ecosystems. Each participant leverages their unique skills and resources to contribute to the overall development of the ecosystem. The free flow of material, energy, and information among participants emerges as a common feature. This seamless sharing enhances collaboration and interdependence, creating a competitive advantage for the entire ecosystem. These shared characteristics collectively lay the foundation for the sustainable development of business ecosystems, innovation ecosystems, and platform ecosystems.

6.Digital Transformation and Technology Integration

The rapid advancement of information and communication technologies, leading to the rise of digital ecosystems, is a common trend across business, innovation, and platform ecosystems. Digital features enhance the flexibility and adaptability of these ecosystems, allowing them to respond effectively to the evolving needs of the business environment.

7.Continuous Innovation and Effective Governance

Notably, James F. Moore’s four-stage development theory proposed in 1993—emergence, expansion, leadership, and self-innovation—applies to these ecosystems, providing a comprehensive blueprint for their common developmental trajectory. Continuous innovation emerges as a vital shared feature, crucial for achieving sustainability and resilience. This enables ecosystems to adapt to unforeseen disturbances and maintain long-term stability. Effective platform governance plays a crucial role in balancing cooperation and competition within the ecosystems, ensuring competitiveness and fostering innovation.

#### 4.2.3. Differentiating the Related Ecosystems

While the BE, IE, and PE share common features, they exhibit distinct characteristics in terms of participant types, core concepts, goals, features, value co-creation, and governance. The BE encompasses a broad range of organizational connections, emphasizing the overall value creation through collaboration and competition, and is distinguished by interdependence among its components. In contrast, the IE involves enterprises, innovators, and research institutions, focusing on innovation and an open environment that encourages multi-party contribution to greater innovation. The PE, comprising providers, developers, partners, and customers, centers on the platform, fostering diversity and an open environment to prompt collaboration and competition dynamics for joint innovation. Each ecosystem emphasizes value co-creation, with the BE fostering overall collaboration, the IE focusing on new ideas and technologies, and the PE allowing customization and creating shared value. Governance mechanisms differ, with the BE requiring a stable organizational structure, the IE needing mechanisms to balance interests, and the PE relying on effective governance for coordination and decision-making. Despite these differences, intersections exist, such as multiple platform ecosystems within a business ecosystem, highlighting diverse aspects of business advancement and innovation across contexts.

## 5. Discussion and Implications

### 5.1. Business Ecosystem

The BE, intricately connected and dynamic, encompasses a diverse array of participants, building a network that extends beyond the supply chain [29]. Focal firms, suppliers, complementors, and customers are central actors in this ecosystem, interconnected at various points upstream and downstream (Figure 7). Suppliers, vital components of the upstream segment, extend beyond mere resource providers, acting as crucial partners collaborating with focal firms. Their contribution goes beyond the supply of materials; through cooperation, they foster innovation, enhance production efficiency, and collectively improve the ecosystem’s competitiveness and efficiency [15,174].

Focal firms, the core of the business ecosystem, transcend the traditional manufacturing scope, including service and platform providers. As organizers and coordinators, focal firms bear the responsibility of creating value [175,176]. Two-way communication mechanisms with suppliers facilitate information exchange, aiding suppliers in understanding market performance and encouraging innovations that enhance the entire ecosystem. The success of focal firms, integral to the ecosystem’s health, relies on effective collaboration with downstream complementors and customers.

Complementors, including government, management institutions, and universities, operate both upstream and downstream. Upstream complementors assist suppliers, augmenting the quality and innovation of products or services. Downstream complementors add value to focal enterprises’ offerings, enhancing user experience and expanding the overall product portfolio of the ecosystem. This complementary relationship fosters synergy, resulting in a more comprehensive solution and elevating the attractiveness of the business ecosystem.

Customers, positioned as end-users, play a pivotal role in the ecosystem. Their needs and feedback are the driving forces for innovation and improvement, significantly influencing the ecosystem’s success. The interaction between customers, focal firms, and complementors directly shapes market performance and the sustainable development of the system.

The intricate relationships among these participants form a highly interdependent and synergistic network. Collaborative efforts and information sharing enhance the system’s flexibility and resilience, enabling it to adeptly respond to market changes and evolving customer needs. This co-evolutionary process not only optimizes internal components but also fuels continuous innovation and development throughout the entire business ecosystem. This complexity invites further research into the dynamics, challenges, and potential for innovation within business ecosystems, exploring how these interdependencies contribute to resilience and sustained development.

### 5.2. Innovation Ecosystem

The IE, characterized by its dynamic and intricate network, operates across multiple levels, with various roles collaborating for collective innovation and value co-creation [7,177]. Three primary entities form the crux of the innovation ecosystem: focal firms, innovators, and innovation support roles (Figure 8); their close relationship and interactive contributions are pivotal for the ecosystem’s success.

Focal firms, serving as the core participants, assume coordination and leadership roles, often being product or service providers with market-oriented and commercial capabilities [178]. Their influence extends beyond their success, with their leadership guiding the direction of innovation, forging partnerships, and managing resources within the ecosystem. The prosperity of the entire ecosystem is intricately linked to the success of focal firms. Through structural, contextual, and coordination capabilities, focal firms foster internal and external linkages, contributing to the collaborative creation and sharing of new value, thereby nurturing sustainable innovation ecosystems.

The innovators, the direct creators of value, encompass various roles such as suppliers, complementary innovators, and users. Suppliers, integral to the ecosystem, collaborate to drive innovation in products and services by providing crucial components, technologies, resources, or feedback [179]. Complementary innovators, including other companies, research institutions, or individuals, collaborate to achieve common innovation goals. Users, participating in product and service use and feedback, contribute essential market information for innovation and occasionally engage in the joint innovation process. These roles form an organic system through cooperation, working collaboratively to create and provide an overall value proposition.

Innovation support roles, comprising experts, universities, research institutes, and other entities, provide peripheral support elements despite not being directly involved in product or service manufacture. Experts and consultants offer domain-specific knowledge and skills, supporting innovation participants with advice and guidance. Universities and research institutions serve as knowledge innovators, advancing scientific frontiers and providing academic research to drive innovation within the ecosystem [102].

The relationships within the innovation ecosystem emphasize synergy and co-creation. The connection between focal firms and innovators forms the core organizational and executional relationship, where the focal enterprise provides a platform and support for innovators, who, in turn, inject new thinking and energy, fostering innovation and development. Focal firms collaborate with innovation supporters, seeking their cooperation and receiving corresponding help and support. Innovators and innovation supporters form a strategic relationship, with innovators proposing new ideas, products, and services and innovation supporters providing essential resources, funds, and expertise. The collaborative efforts of these roles are crucial for achieving common value-creation goals and sharing the benefits brought by the success of innovation, thereby ensuring the long-term health and sustainable development of the innovation ecosystem [180]. This intricate interplay of roles and relationships presents avenues for further research into understanding and optimizing the dynamics of innovation ecosystems.

### 5.3. Platform Ecosystem

The platform ecosystem, depicted in Figure 9, is a dynamic interplay of three core entities: platform providers, complementors, and end-users, each contributing uniquely to the ecosystem’s evolution [181]. At the heart of this ecosystem is the platform provider, responsible for establishing the technical infrastructure, defining interaction rules, and shaping the conditions for collaboration. The symbiotic relationship between platform providers and complementors is fundamental, based on non-shrinkable product complementarities. The platform provider furnishes a robust infrastructure, enabling complementors to innovate and develop products, applications, features, or services that complement the original offerings, enhancing the overall value for end-users [71]. This dynamic interaction continually reshapes the user value of the platform, influencing the contributions of other complementors within the ecosystem.

Complementors, ranging from developers to service providers, thrive on the open and collaborative nature of the platform, actively participating in value co-creation and innovation. This diversity fosters a multifaceted ecosystem with varied skills and resources, contributing to its dynamism. The continuous feedback loop between complementors and end-users is coordinated through market-based mechanisms, with end-users providing valuable insights, needs, and preferences that guide complementors to enhance products and services. This iterative process ensures a user-centric ecosystem that is responsive to evolving needs. However, despite pursuing a common value-creation goal, complementors in the platform ecosystem do not sign cooperation agreements with each other [182,183].

End-users, as ultimate consumers, actively participate in the PE, providing feedback, generating data, and even engaging in collaborative innovation. This active involvement transforms users into co-creators, contributing to product/service development and increasing the overall value of the PE [30]. The mutual-benefit relationship between end-users and platform providers involves customization of the platform to meet user needs, ensuring user satisfaction and loyalty. End-users serve as a critical link for complementors, providing feedback and preferences that guide the creation of applications and services, fostering a user-centered ecosystem driven by continuous communication.

To recapitulate, PE is a network of close interactions involving platform providers, complementors, and end-users. Platform providers dictate the rules and create a symbiotic relationship with complementors, who, in turn, enrich the ecosystem through innovation, forming a feedback loop with end-users. The participatory role of end-users goes beyond consumption, actively contributing to the co-creation of value. This collaborative dynamic among the three entities propels innovation, diversity, and user value within the platform ecosystem.

### 5.4. Future Research

To shape the direction of future research on ecosystems, we systematically reviewed 28 pertinent documents from the literature spanning the years 2020 to 2023. Our selection process involved a meticulous assessment of each document’s alignment with the three core themes: business ecosystem, innovation ecosystem, and platform ecosystem. The overarching goal of our analysis was to reveal the research objectives and significant findings associated with each theme. A comprehensive summary of the selected literature, delving into the distinct focus of each study and encapsulating its notable contributions, is presented in detail in Table 6.

After conducting an extensive analysis of the recent literature, we have identified cutting-edge research topics relating to BEs that have gained prominence. Given the escalating unpredictability and randomness of today’s social environment, organizational structures and societal ecosystems are increasingly vulnerable to unforeseen and disruptive events [184,185]. Consequently, research into the resilience and recovery of business ecosystems has emerged as a critical area of study. Ramezani and Camarinha-Matos (2020) delved into the concepts of business ecosystem resilience and anti-vulnerability, categorizing sources and drivers of disruption and proposing coping strategies [118]. Burford et al.’s (2022) exploration focused on the relationship between component selection and performance in business ecosystems post-negative shocks, offering guidance for ecosystem construction [127]. Masucci et al. (2020) emphasized enhancing the competitiveness of business ecosystems through coordinated open innovation strategies [114]. Sustainable development is another significant theme, with Lee and Roh (2023) studying the role of cooperation strategy and digital capability in achieving sustainability [126]. Sun et al. (2020) conducted a detailed study on the construction mechanism of “Internet+ WEEE collection”, a commercial ecosystem of renewable resources, highlighting the pivotal role of digital technology in business ecosystem construction and development [132]. Digital technology’s role in business ecosystems has been extensively researched [128,129,130,131].

In recent research on IEs, the theme of digitization has garnered considerable attention. Linde et al. (2021) examined how firms should develop dynamic capabilities to navigate digital opportunities in dynamic ecosystem environments [108]. Beltagui et al. (2020) emphasized the disruptive potential of digital innovation ecosystems [103]. Kamalaldin et al. (2021) focused on configuring ecosystem strategies for equipment suppliers to enable digital process innovation in process industry companies [113]. Elia et al. (2020) investigated the impact of digital technology on the entrepreneurial process, proposing the concept of a digital entrepreneurial ecosystem [96]. Stahl’s (2022) research centered on establishing a responsible AI innovation ecosystem [106]. Beyond digitalization, new orientations in IEs, such as providing Industry 4.0 solutions and intelligent product innovation, have also become areas of interest for future researchers [104,105,111]. Resilience in IEs has also been a subject of exploration, echoing trends observed in business ecosystems. Liang and Li (2023) highlighted the crucial role of regional IE resilience, indicating its positive spillover effect on the digital economy’s development [107]. Sustainable development goals have drawn extensive attention, with studies discussing factors contributing to the stable development of green IEs and exploring the impact of relevant regulations on collaborative innovation from a policy perspective [110]. Furthermore, researchers have discussed the role of transnational corporations in realizing sustainable development goals [112].

In the realm of PEs, recent research has prominently featured the theme of digitization. Jovanovic et al. (2022) conducted an in-depth study on the evolution process and mechanisms of industrial digital platforms [82]. Sandberg et al.’s research (2020) explored the crucial role of digital capabilities in the development and phase transition of product platforms [92]. Hilbolling et al. (2021) focused on the quality of complementary products in the digital PE [86]. Schreieck et al.’s (2021) paper identified critical capabilities for building digital PEs and their role in facilitating value co-creation and mechanism acquisition [91]. Similar to BEs and IEs, “resilience” has gained recent research attention in PE. Floetgen et al. (2021) conducted an in-depth discussion on the resilience of PEs in the context of COVID-19 [89]. Additionally, there are topics that deserve further research, such as guidance for enterprises dealing with dominant platforms affecting their competitiveness [87] and the impact of user preferences and demand-based economies of scale on PE dynamics [90].

Based on our interpretation of research themes in business, innovation, and platform ecosystems over the past three years, we have outlined relevant topics and directions that have garnered significant attention. Regardless of the ecosystem type, integrating with digital technology, achieving sustainable development, and enhancing resilience and anti-vulnerability are ongoing research topics that warrant continuous attention. We recommend that future researchers and practitioners delve deeply into these aspects to contribute to the advancement of these fields.

## 6. Conclusions

### 6.1. Contributions

Our research methodology combines bibliometric analysis and content analysis, offering a comprehensive approach to systematically organize and understand trends and concepts in BEs, IEs, and PEs. Specifically, this paper utilizes CiteSpace to conduct several key bibliometric analyses on literature samples from the WoS database, visually exploring the current state of development and evolving trends in these fields. Simultaneously, through the content analysis of 32 authoritative and highly cited articles, we establish a clear definition framework and summarize the core concepts and characteristics of these ecosystems. This analysis assists researchers and practitioners in distinguishing between these three ecosystems, enabling them to select models that meet their specific needs.

After analyzing influential articles on BEs, IEs, and PEs, their definitions can be distilled. The BE is a dynamic network involving various players beyond the supply chain to create value and achieve the focal firm’s strategic goals. The IE is an intricate network where focal firms, innovators, and innovation supporters collaborate for collective innovation and value co-creation. The PE is a collaborative network centered on platforms, where platform providers establish infrastructure and collaboration rules, complementors offer supplementary products, and end-users provide data, each contributing uniquely to the ecosystem’s evolution. Biomimicry serves as an innovation tool that harnesses design principles from nature to foster innovative thinking [186]. For example, in business ecosystems, biomimicry principles can inspire companies to optimize resource utilization, develop symbiotic relationships with stakeholders, and create more resilient organizational structures. In innovation ecosystems, biomimicry can foster creativity, guide experimentation, and leverage diverse perspectives to drive breakthrough innovations. In platform ecosystems, biomimicry can inform the design of scalable infrastructures that facilitate value exchange and collaboration among diverse participants. Overall, integrating biomimicry principles into ecosystem development can enhance sustainability, innovation, and resilience [187].

There are both commonalities and differences among BEs, IEs, and PEs concerning participant types, objectives, and value co-creation processes (Figure 10).

8.Types of Participants

In biomimetics, diverse communities of organisms form an interdependent system [188]. Similarly, all three ecosystems involve a diverse set of players who collectively drive their growth and prosperity. However, BEs primarily involve traditional business entities and supply chain collaborations, IEs focus on innovators and researchers, and PEs revolve around platform providers and ecosystem expansion. Nevertheless, akin to individual species in a biological ecosystem, each member of the ecosystem ultimately shares the fate of the entire network [29].

9.Goals

The overarching goals of the three ecosystems involve fostering innovation, enhancing competitiveness, and achieving sustainable growth through collaboration, akin to the pursuit of organisms in biomimicry. Organisms maintain ecological balance and promote survival and prosperity through interdependence and interaction. In BE, the focus is on optimizing supply chain efficiency and market competitiveness. IE drives technological advancement and disruptive innovation, emphasizing open collaboration, while PE aims to create scalable platforms and promote ecosystem growth.

10.Value Co-Creation

Value co-creation serves as the cornerstone of all ecosystems, highlighting the collective creation and sharing of value among participants to foster ecosystem development [95]. This concept mirrors the interdependent and value-co-creating nature of organisms in natural ecosystems, contributing to their stable development. While all ecosystems emphasize value co-creation, their focuses and mechanisms vary. The BE prioritizes synergies in business models, value chains, and markets, emphasizing efficiency, cooperation, and competition within the supply chain. The IE emphasizes collaborative innovation and knowledge sharing, centering on the innovation process involving innovators, supporters, and users, with a focus on developing new products and services [37]. In BEs and IEs, the former acquires value while the latter primarily focuses on value creation. Conversely, the PE prioritizes platform openness, cooperation, and value co-creation among participants, emphasizing platform construction, expansion, and innovation.

Examining the recent trends in research on BEs, IEs, and PEs, we observe a notable focus on the theme of digitization [91,96,126,128]. Simultaneously, resilience emerges as a future research hotspot for both the BE and the IE [107,118]. Moreover, new directions in the innovation ecosystem, such as providing Industry 4.0 solutions and innovating smart products, also garner attention for future research [104,111]. By leveraging insights from research on business, innovation, and platform ecosystems, biomimetic research can better grasp the dynamics and evolution of ecosystems, particularly amidst digital transformation [189]. Conversely, the experience gained from digital transformation can offer methodologies for biomimicry to effectively utilize advanced technologies in simulating and understanding ecosystem dynamics. This interdisciplinary collaboration and mutual inspiration are poised to propel the advancement of biomimicry and offer a more comprehensive perspective for interpreting and modeling natural systems.

Our research delves into the complex dynamics and collaborative behaviors within these ecosystems, offering valuable insights for both academic study and practical application. We emphasize the significant role of BEs, IEs, and PEs in shaping modern business strategy through the lens of biomimicry. By drawing inspiration from the resilience, efficiency, and collaborative dynamics of natural systems, biomimicry principles can drive organizational adaptation and growth. In the realm of the BE, we stress the importance of forging partnerships, optimizing resource utilization, and fostering symbiotic relationships with stakeholders. Similarly, in the IE, we highlight strategies for fostering creativity, nurturing experimentation, and leveraging diverse perspectives to drive innovation. Exploring PEs unveils opportunities to create scalable infrastructures facilitating value exchange and collaboration among diverse participants. Ultimately, our investigation contributes to both academic knowledge and practical implementation, providing actionable insights that inform sustainable and resilient business strategies inspired by nature’s wisdom.

### 6.2. Limitations

The scope of the research outlined in this paper is subject to several limitations that warrant acknowledgment. Firstly, there are constraints related to the sources of literature, as the study heavily relies on the Web of Science (WoS) database. While WoS is well-suited to large-scale bibliometric analyses and enhances relevance to the research topic through categorization, it may not cover all pertinent research. This limitation raises the possibility that relevant literature may have been omitted, potentially leading to deviations in the results. Secondly, the limitations of the retrieval strategy may have resulted in the oversight of important articles that are not captured by the search formula, given its primary focus on the title, abstract, and keyword information of the articles. Additionally, both the bibliometric and content analyses center on the collected literature samples, potentially overlooking other crucial works.

We recognize that our manuscript does not claim to provide the ultimate truth but, rather, offers insights based on our perspectives and understanding of ecosystems in business strategy. Additionally, we advocate for integrating biomimicry within these ecosystems, as it can offer innovative solutions inspired by nature’s designs and principles. Several considerations could be made to enhance the robustness of future research in this domain. Firstly, researchers could integrate other databases, such as Google Scholar, Scopus, etc., to collect samples. This approach would allow for the analysis of literature from multiple databases, providing a more comprehensive and detailed perspective. Secondly, there is room for improvement in the search strategy to include more comprehensive literature samples, ensuring a more exhaustive representation of the relevant literature. Thirdly, integrating additional manual screening processes during sample selection could enhance the effectiveness of the samples while maximizing comprehensiveness. Finally, regular data updates should be considered to capture the latest research trends, ensuring the timeliness and relevance of the research results. Addressing these considerations in future research endeavors will contribute towards overcoming the identified limitations and further advance our understanding of business, innovation, and platform ecosystems.

## Figures and Tables

**Figure 1 biomimetics-09-00216-f001:**
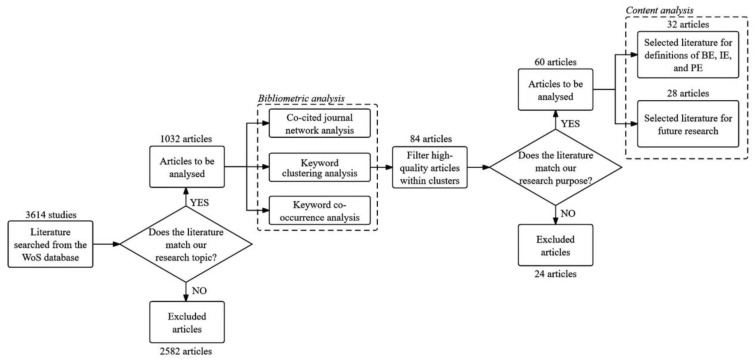
Stages of the systematic review.

**Figure 2 biomimetics-09-00216-f002:**
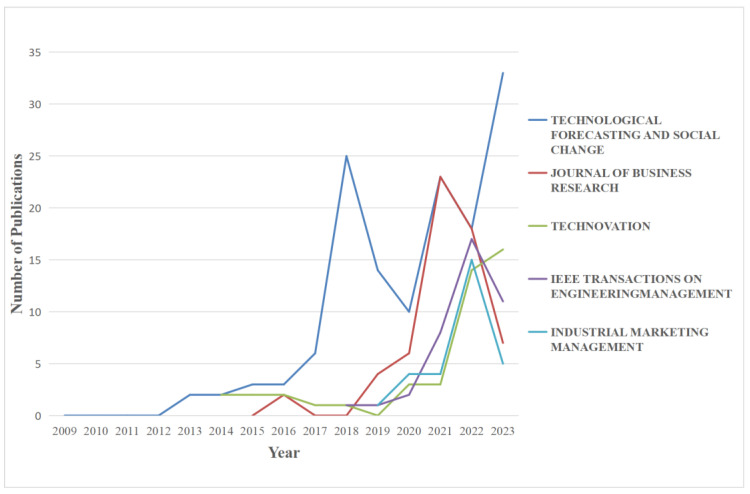
Annual publications of the top five popular journals.

**Figure 3 biomimetics-09-00216-f003:**
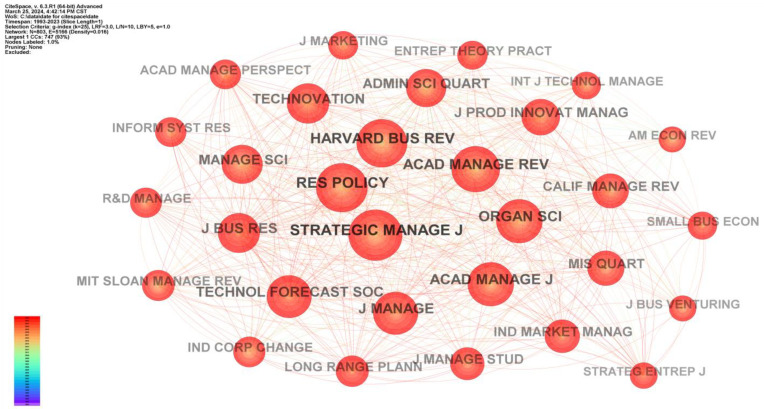
The co-citation network of journal publications.

**Figure 4 biomimetics-09-00216-f004:**
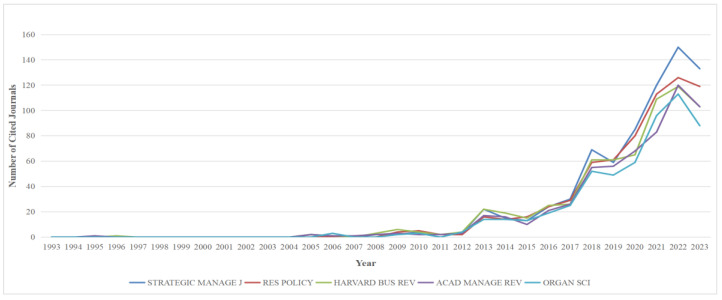
Co-citation trends in top five journals (1993–2023).

**Figure 5 biomimetics-09-00216-f005:**
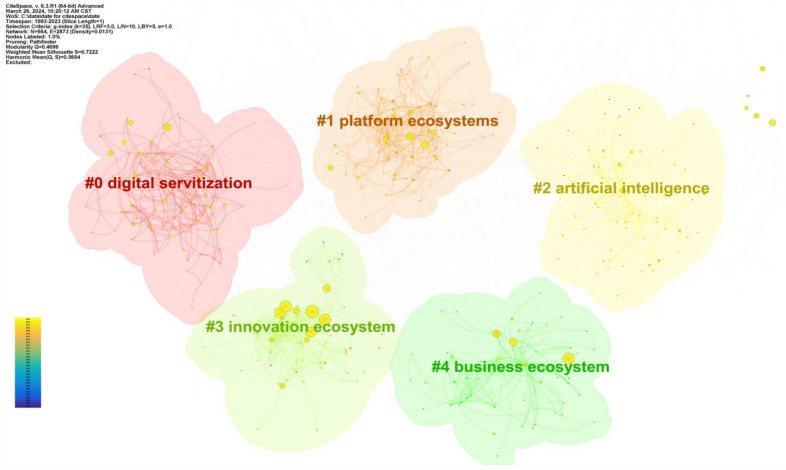
Keyword cluster analysis.

**Figure 6 biomimetics-09-00216-f006:**
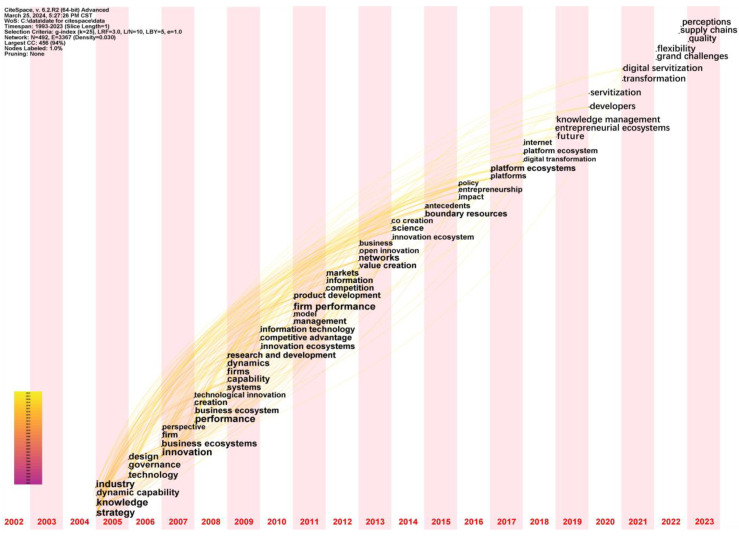
The keyword time-zone of publications in 2004~2023.

**Figure 7 biomimetics-09-00216-f007:**
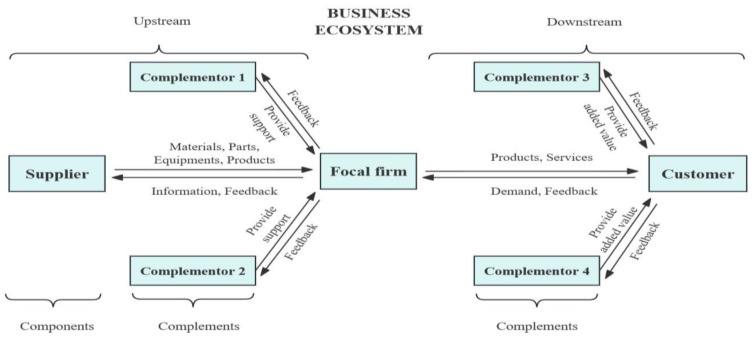
The relationship between the four themes within the BE.

**Figure 8 biomimetics-09-00216-f008:**
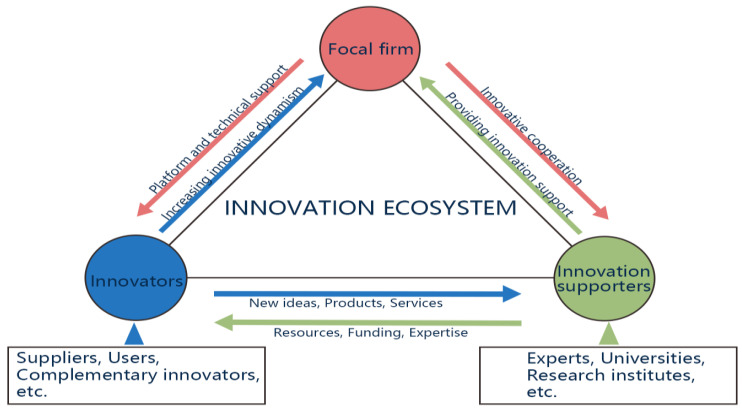
The relationship between the three themes within the IE.

**Figure 9 biomimetics-09-00216-f009:**
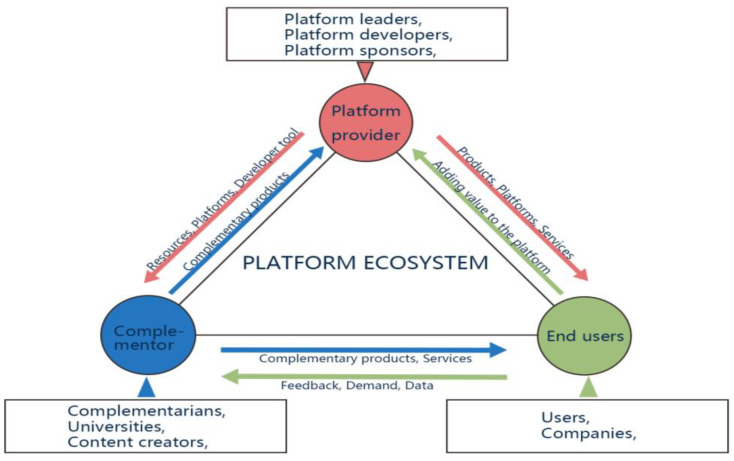
The relationship between the three themes within the PE.

**Figure 10 biomimetics-09-00216-f010:**
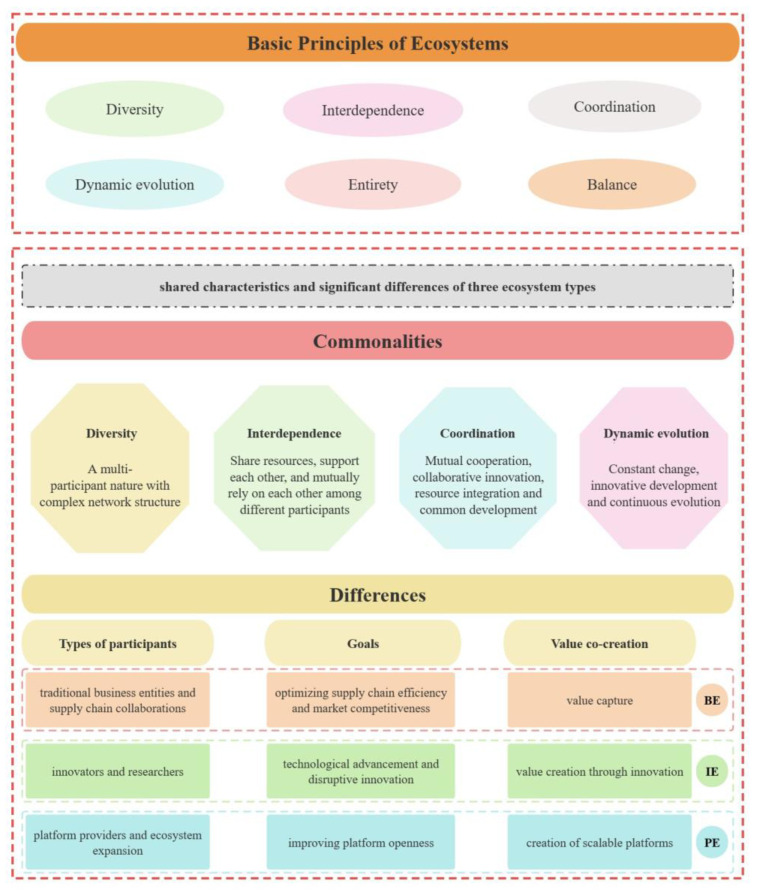
Core principles, key commonalities and differences among three ecosystem types.

**Table 1 biomimetics-09-00216-t001:** Examples of coding.

Definition	Element	Role	Source	Literature
Business ecosystem (BE)	Focal firm	Product or service maker	The BE has a loose network of suppliers, distributors, outsourcing firms, makers of related products or services, technology providers, and a host of other organizations.	[39]
		Architect	A focal firm needs not only to develop linkages with its potential direct partners but also to create an entire ecosystem involving indirect partners.	[54]
	Supplier	Provider	…interdependent stakeholders, encompassing users, rivals, providers, community groups, and various entities…	[55]
	Complementor	Complementary partner	Different stakeholders include direct industrial players, government agencies, industry associations, competitors, and customers.	[54,56]
	Customer	Demand	The main difference between business and innovation ecosystems seems to be a lack on the demand side (customer/ user) in the latter.	[57]
	Principles	Diversity	…the combination of all the efforts of all players of the community (large and small–medium manufacturers, retailers, government, technological parks, universities, consultants, etc.) guarantees the survival and the success of the BE.	[58,59]
		Interdependence	…highlights the interdependence of all actors in the business environment, who co-evolve their capabilities and roles.	[14,60,61]
		Coordination	…showing a business ecosystem which is seen as a class of CN (collaborative network), more specifically as a sub-class of a long-term strategic network.	[12,61]
		Dynamic evolution	In a business ecosystem, companies co-evolve capabilities around a new innovation: they work cooperatively and competitively to support new products, satisfy customer needs, and eventually incorporate the next round of innovations.	[6,62,63]
Innovation ecosystem (IE)	Focal firm	Advocate	The main driver is the differences in innovation incentives and strategies among the different focal firms.	[15,64]
	Innovators	Supplier	Underlying a technology’s advance are not only efforts by producers of the focal technology but also systemic efforts by component and complement providers from a range of interdependent industries.	[64]
		Complementary innovator	…leading to a dynamic innovation ecosystem in which complementary innovators were continuously providing new value to customers…	[16]
		User	Developing and modifying new products with users; adopting innovative solutions provided by users.	[65]
	Innovation supporters	Expert	This role of ‘entrepreneur’ may be assumed in response to the partnership-forging activities of the ecosystem leader or as a result of seeing opportunities to commercialize the discoveries and inventions of experts.	[17]
		Research institute	Through cooperation with universities and research institutes, the firm acquires and utilizes innovative resources and commercializes cooperative R&D results to promote innovation.	[65]
	Principles	Diversity	The innovation ecosystem is composed of interconnected and interdependent networked actors, which include the focal firm, customers, suppliers, complementary innovators, and other agents as regulators.	[37]
		Interdependence	There is an understanding that both BEs and IEs are composed of interconnected and interdependent network actors.	[37,66]
		Coordination	An innovation ecosystem is set for the co-creation or the joint creation of value.	[37,67]
		Dynamic evolution	The evolving relationships between the wide range of innovation partners in an innovation ecosystem highlight the degree to which their interaction contributes to knowledge creation…	[68]
Platform ecosystem (PE)	Platform provider	Sponsor	Platform leaders craft governance mechanisms that are inherent in the platform’s technical architecture.	[20,69,70]
	Complementor	Complementary partner	Platform owners often seek to encourage complementary third-party innovation from sources external to the organization, including customers, research firms, business partners, and universities.	[19,21]
	End users	Consumer	The platform leader, complementors, and users who consume these products or services collectively comprise the ecosystem surrounding the platform.	[20]
		Company	Startups’ technology commercialization can be facilitated by joining a platform ecosystem.	[21]
	Principles	Diversity	The platform ecosystem exhibits a diversity of ownership and control, of both complementary assets and the components that make up the platform.	[18]
		Interdependence	…through their participation in the ecosystem, complementors constantly reshape the platform’s user value through the variety of complements they create and, in doing so, also affect the value for other complementors to participate in the ecosystem.	[71]
		Coordination	The platform can leverage these relationships to foster, for instance, tighter collaboration and provide “rewards” …	[72,73]
		Dynamic evolution	Platform ecosystems have been discussed as complex ecologies of firms with individual and collective intertwined interests, whose evolution follows some emergent self-organizing patterns based on complementarities and the coevolution of participants’ activities and capabilities.	[71,72]
		Balance	The platform owner should strive for a balance between incremental and radical innovation by complementors as well as the success of complementors’ innovation in terms of the performance of goods.	[18]

**Table 2 biomimetics-09-00216-t002:** Topics, keywords, authors, and journals.

Topic	Keyword	Author(s)	Journal	
#0 Digital servitization	Digital servitization Internet Opportunity Organizations Servitization	Jovanovic et al., 2022 [82]	1	*Technovation*
		Sjödin et al., 2022 [83]	2	*California Management Review*
		Kohtamaki et al., 2019 [84]	3	*Journal of Business Research*
		Sklyar et al., 2019 [85]	4	*Journal of Business Research*
#1 Platform ecosystems	Competition Impact Business Architecture Digital transformation Platform ecosystem Boundary resources Markets	Hilbolling et al., 2021 [86]	5	*Journal of Product Innovation Management*
		Ceccagnoli et al., 2012 [21]	6	*Mis Quarterly*
		Schmeiss et al., 2019 [20]	7	*California Management Review*
		Khanagha et al., 2022 [87]	8	*Strategic Management Journal*
		Cennamo et al., 2019 [71]	9	*Organization Science*
		Tavalaei and Cennamo, 2021 [72]	10	*Long Range Planning*
		Inoue, 2021 [73]	11	*Technological Forecasting and Social Change*
		Gawer and Cusumano, 2014 [88]	12	*Journal of Product Innovation Management*
		Floetgen et al., 2021 [89]	13	*European Journal of Information Systems*
		Murthy and Madhok, 2021 [19]	14	*Journal of Management Studies*
		Kretschmer et al., 2022 [69]	15	*Strategic Management Journal*
		Panico and Cennamo, 2022 [90]	16	*Strategic Management Journal*
		Schreieck et al., 2021 [91]	17	*Journal of Information Technology*
		Sandberg et al., 2020 [92]	18	*Management Information Systems Quarterly*
		Thomas et al., 2014 [18]	19	*Academy of management perspectives*
		Cenamor and Frishammar, 2021 [70]	20	*Research Policy*
#2 Artificial intelligence	Business models Perspective Science Knowledge management Big data	Burström et al., 2021 [93]	21	*Journal of Business Research*
		Leone et al., 2021 [94]	22	*Journal of Business Research*
		Manser et al., 2021 [95]	23	*Journal of Research in Interactive Marketing*
		Elia et al., 2020 [96]	24	*Technological Forecasting and Social Change*
		Ehret and Wirtz, 2017 [97]	25	*Journal of Marketing Management*
		Clough and Wu, 2022 [98]	26	*Academy of Management Review*
#3 Innovation ecosystem	Innovation Performance Strategy Technology Knowledge Innovation ecosystem Framework Platforms Creation Design	Walrave et al., 2018 [66]	27	*Technological Forecasting & Social Change*
		Mollenkopf et al., 2021 [99]	28	*Journal of Service Management*
		Bart Clarysse et al., 2014 [58]	29	*Research Policy*
		Granstrand and Holgersson, 2020 [16]	30	*Technovation*
		Adner and Kapoor, 2016 [64]	31	*Strategic Management Journal*
		Robertson et al., 2021 [68]	32	*International Business Review*
		Oh et al., 2016 [100]	33	*Technovation*
		Dedehayir et al., 2018 [17]	34	*Technological Forecasting and Social Change*
		Wagner, 2021 [101]	35	*International Journal of Physical Distribution & Logistics Management*
		Ben Letaifa, 2014 [102]	36	*Management Decision*
		de Vasconcelos Gomes et al., 2018 [37]	37	*Technological Forecasting and Social Change*
		Beltagui et al., 2020 [103]	38	*Research policy*
		Kahle et al., 2020 [104]	39	*Technological Forecasting and Social Change*
		Benitez et al., 2020 [105]	40	*International Journal of Production Economics*
		Stahl, 2022 [106]	41	*International Journal of Information Management*
		Liang and Li, 2023 [107]	42	*Technological Forecasting and Social Change*
		Linde et al., 2021 [108]	43	*Technological Forecasting and Social Change*
		Audretsch et al., 2022 [109]	44	*International Entrepreneurship and Management Journal*
		Yang et al., 2021 [110]	45	*Journal of Cleaner Production*
		Yin et al., 2020 [111]	46	*Journal of Cleaner Production*
		Nylund al., 2021 [112]	47	*Journal of Cleaner Production*
		Kamalaldin et al., 2021 [113]	48	*Technovation*
		Shaw and Allen, 2018 [67]	49	*Technological Forecasting and Social Change*
		Ander and Kapoor, 2010 [15]	50	*Strategic Management Journal*
		Xie and Wang, 2020 [65]	51	*Journal of Business Research*
		Masucci et al., 2020 [114]	52	*Research Policy*
		Rohrbeck et al., 2009 [115]	53	*R & D Management*
		Radziwon and Bogers, 2019 [116]	54	*Technological Forecasting and Social Change*
		Chesbrough et al., 2014 [117]	55	*California Management Review*
#4 Business ecosystem	Value creation Management Model Business ecosystem Evolution Firm performance Competitive advantage Business model	Moore, 1993 [6]	56	*Harvard Business Review*
		Ramezani and Camarinha-Matos, 2020 [118]	57	*Technological Forecasting & Social Change*
		Gupta et al., 2019 [60]	58	*Technological Forecasting & Social Change*
		Riquelme-Medina et al., 2022 [119]	59	*Journal of Business Research*
		Audretsch et al., 2019 [120]	60	*Journal of Technology Transfer*
		Ketchen Jr et al., 2014 [121]	61	*Journal of Business Logistics*
		Best, 2015 [122]	62	*Technovation*
		Scaringella and Radziwon, 2018 [57]	63	*Technological Forecasting and Social Change*
		Hakala et al., 2020 [62]	64	*International Journal of Management Reviews*
		Rong et al., 2015 [54]	65	*Journal of International Management*
		Graça and Camarinha-Matos, 2017 [61]	66	*Technological Forecasting and Social Change*
		Song, 2019 [123]	67	*Small Business Economics*
		Nieuwenhuis et al., 2018 [124]	68	*Technological Forecasting and Social Change*
		Bals, 2019 [125]	69	*Journal of Purchasing and Supply Management*
		Kapoor and Li, 2013 [56]	70	*Strategic Management Journal*
		Tsujimoto et al., 2018 [12]	71	*Technological Forecasting and Social Change*
		Li, 2009 [39]	72	*Technovation*
		Pierce, 2009 [14]	73	*Strategic Management Journal*
		Battistella et al., 2013 [59]	74	*Technological Forecasting and Social Change*
		Ma et al., 2018 [63]	75	*Journal of Cleaner Production*
		Lee and Roh, 2023 [126]	76	*Journal of Cleaner Production*
		Burford et al., 2022 [127]	77	*Strategic Management Journal*
		Hanelt et al., 2021 [128]	78	*Journal of Management Studies*
		Palmié et al., 2022 [129]	79	*Technological Forecasting and Social Change*
		Battisti et al., 2022 [130]	80	*Technological Forecasting and Social Change*
		Chen et al., 2023 [131]	81	*The Journal of Technology Transfer*
		Sun et al., 2020 [132]	82	*Journal of Cleaner Production*
		Sun et al., 2018 [133]	83	*Journal of Cleaner Production*
		Yi et al., 2022 [55]	84	*Technovation*

**Table 3 biomimetics-09-00216-t003:** Business ecosystem (BE).

Literature	Definition of BE	Research Objective	Methodology	Research Finding(s)
Moore (1993) [6]	BEs, in contrast to co-evolving organisms in biological communities, are social systems sustained through an intricate network of choices made by participants.	Dominant company	Metaphor method	A BE comprises stages of birth, expansion, leadership, and self-renewal.
Li (2009) [39]	A BE is a group of organizations such as suppliers, distributors, manufacturers, and tech providers. They are connected through platforms, making their collaborative efforts more valuable than what each can achieve alone.	Business ecosystem M&A strategy	Case study	The M&A strategy provides an accelerated approach to complement the company’s core technology portfolio.
Pierce (2009) [14]	BEs occupy a continuous space consisting of closely related suppliers, customers, complementors, and core companies. Together, they shape the technological architecture for creating products and services.	The impact of core enterprise decision-making on complementary enterprise performance	Quantitative analysis	The dynamic product design strategies of major enterprises and the entrance of niche players have generated turbulent ecosystems, leading to financial setbacks and exits for independent niche market firms.
Kapoor and Lee (2013) [56]	BE is a complex network composed of companies and their customers, complementors, and suppliers with interdependent relationships.	The impact of organizational differences in complementary aspects of enterprises on investment in new technologies within the BE	Quantitative analysis	In addition to affecting incentives and costs, the way a firm and its complementors are organized plays a crucial role in the firm’s ability to coordinate changes related to complementary activities. This coordination is key to reaping the benefits of early investments in new technologies.
Battistella et al. (2013) [59]	A complex system with multiple interconnected loops, both within and between them, featuring mutual cross-feed relationships and inhibitory connections. It also involves preferential reactions depending on varying substrate concentrations.	The arrangement and flow within a BE	Case study	The initial proposition of a systematic approach for examining both the static and dynamic structures of a business ecosystem is called the methodology of business ecosystem network analysis (MOBENA).
Clarysse et al. (2014) [58]	BE can be viewed as a collective of companies collaborating to create value by leveraging their skills and assets concurrently.	The connection between knowledge ecosystems and BEs	Social network analysis	In knowledge ecosystems, there is a tendency to concentration around a few central actors, while business ecosystems exhibit limited presence at the local level. This highlights fundamental distinctions in the processes of value creation between the two.
Rong et al. (2015) [54]	The BE functions as a self-contained economic community comprising diverse stakeholders, such as direct industry participants, government agencies, industry associations, competitors, and customers. These entities mutually benefit from each other and share similar outcomes.	Cultivating BE for enterprises	Case study	To develop a BE in a new foreign market, go through three stages: begin by nurturing complementary partners, then identify leadership partners, and finally integrate ecosystem partners. Key activities involve sharing vision, identifying leaders, and connecting partners throughout this process.
Graça and Camarinha-Matos (2017) [61]	BE is an economic community made up of interacting organizations and individuals.	An indicator system for evaluating performance in a collaborative business ecosystem	Literature review	The digital BE can be comprehended as consisting of digitization, commerce, and ecosystem, with attributes such as economy, commerce, population, community, multi-agent system, dynamics, evolution, and network.
Scaringella and Radziwon (2018) [57]	BE is described as a network of closely related companies, either interconnected or situated in close geographical proximity.	The invariant terminology used in various studies of ecosystems	Literature review	Innovation, value acquisition, and competitive advantage are the goals of the business ecosystem.
Ma et al. (2018) [63]	BE is an economic community comprising diverse stakeholders, such as industry participants, governments, industry associations, competitors, customers, and others, coexisting in the same economic landscape and evolving collectively.	The interplay between social-ecological innovation and the sustainable development of urban systems within the sharing economy	Case study	A robust co-evolution mechanism exists between the macro-level transformation towards a more sustainable city and the meso-level innovation in the BE, particularly in the development of greener and smarter transport.
Tsujimoto et al. (2018) [12]	BE is an organizational structure centered on value acquisition and creation, composed of a business environment and numerous private enterprises.	The research direction of ecosystems within technology management and innovation	Literature review	Four major research streams include an industrial ecology lens, a business ecosystem viewpoint, platform management, and a multi-actor network approach.
Gupta et al. (2019) [60]	The fundamental concept of a BE revolves around three keywords: business participants, network, and strategy.	Boundaries of BEs, IEs, and digital ecosystems	Bibliometric analysis	A BE typically emphasizes actors, networks, and strategies.
Hakala et al. (2020) [62]	The key themes of the BE are internationalized, worldwide rivalry, and collaboration, moving towards collaborative competition and co-evolution within the interconnected system.	Concept of ecosystem terminology	Literature review	In the BE, only companies that adapt to the environment, innovate continuously, and collaborate for mutual benefit can survive and achieve success.
Yi et al. (2022) [55]	BEs consist of interdependent stakeholders, encompassing users, rivals, providers, community groups, and various entities, along with the relationships among them.	The influence of interactions with stakeholders on the business model innovation of focal enterprises	Quantitative analysis	The association among industry stakeholders exhibits an inverted U-shaped correlation with business model innovation, whereas the connection with stakeholders outside the industry positively influences business model innovation.

**Table 4 biomimetics-09-00216-t004:** Innovation ecosystem (IE).

Literature	Definition of IE	Research Objective	Methodology	Research Finding(s)
Ander and Kapoor (2010) [15]	The IE consists of the focal actors and the external environment (upstream components and downstream complements) with interdependence.	External innovation challenges	Case study	A universal structure of an ecosystem comprises upstream components, focal actors, downstream complements, and customers. Challenges in upstream innovation will strengthen the leading edge of focal actors, while challenges in downstream innovation will weaken it.
Granstrand and Holgersson (2020) [16]	IE is the evolving combination of participants, actions, elements, establishments, connections, and mutually supportive elements. It is essential for the innovative performance of an individual actor or a population of actors.	Concept of IE	Literature review	Participants, artifacts, and activities constitute the components of the IE, with elements interconnected through complementary and substitutive relationships. The IE exhibits a characteristic of continuous development.
Ander and Kapoor (2016) [64]	IEs are made up of interdependent constituents and supplements, within which essential technologies are embedded.	Key technologies	Case study	The rate at which the new technology replaces the current technology will hinge on the combined levels of the ecosystem emergence challenge for the new technology and the ecosystem expansion opportunity for the existing technology.
Dedehayir et al. (2018) [17]	IE describes the collaborative efforts of different actors for innovation and follows the evolution of the four stages of the ecosystem life cycle—creation, growth, dominance, and revitalization.	IE actors	Literature review	Based on the specific activities of the actors in the birth stage of the IE, participants are categorized into four roles: guiding roles, direct value creation roles, supporting roles, and contributing roles. The roles played by actors will shift as the IE develops, and participants enter or exit at different times during the birth stage, affecting the dynamics of the IE.
Walrave et al. (2018) [66]	IE functions as a network of interdependent actors who pool specialized yet complementary resources and capabilities to collaboratively co-create and deliver a comprehensive value proposition to end-users while also appropriating the gains obtained in the process.	Path-breaking innovation	Literature review	Four approaches are suggested to enhance external viability: (1) develop the value proposition and ecosystem model by learning from socio-technical experimentation feedback; (2) learn from organizations pioneering innovations to shape the value proposition and ecosystem model; (3) align the value proposition and ecosystem model with the evolving development trajectory in socio-technical niches; (4) implement niche protection schemes and maintain sufficient resource slack.
de Vasconcelos Gomes et al. (2018) [37]	IE takes co-creation or joint creation of value as its main goal and follows the process of ecosystem co-evolution. It is composed of interrelated and interdependent participants who face cooperation and competition.	Concept of IE	Bibliometric, content analysis	Like BEs, IEs involve interconnected players led by key figures or platform leaders. Participants in IEs engage in both cooperation and competition, driving co-evolution. However, IEs focus on value co-creation, while BEs prioritize value capture.
Xie and Wang (2020) [65]	IE is a loosely interlinked community of companies and other entities that coevolve strengths around a shared set of technologies, insights, or expertise. They collaborate and compete to innovate new products and services.	Modes of open IE	Grounded theory, FSQCA (fuzzy set qualitative comparative analysis)	Relying only on isolated open innovation is insufficient for improving enterprise product innovation. To enhance innovation capabilities, consider three model combinations involving interfirm cooperation, firm-intermediary collaboration, technology transfer, and collaborative efforts with other entities.
Robertson et al. (2021) [68]	IE is an evolving collection of participants, activities, and artifacts with complementary and substitutive relationships. It delivers value by facilitating the exchange of information and ensuring access to resources, all grounded in knowledge and related practices to achieve innovative results.	Innovation performance	Partial least squares path analysis	Knowledge creation strongly influences innovation performance in advanced and emerging economies. Knowledge diffusion is crucial for emerging markets, while knowledge absorption is key for transitioning economies. Knowledge impact is also vital for innovation performance in transitioning and emerging economies.
Shaw and Allen (2018) [67]	IEs are interconnected business model pathways.	Natural ecosystems	Case Study	IEs involve the serial recycling of scarce resources through ecosystem pathways.

**Table 5 biomimetics-09-00216-t005:** Platform ecosystem (PE).

Literature	Definition of PE	Research Objective	Methodology	Research Finding(s)
Ceccagnoli et al. (2012) [21]	PEs are comprised of interconnected technology platform owners collaborating with other companies to generate business value.	Small independent software developers	Quantitative analysis	Complementary innovation networks make platforms more valuable.
Schmeiss et al. (2019) [20]	PEs encompass leaders responsible for designing and overseeing technical architectures, as well as collaborators and consumers of the offered products or services.	Building a new PE for startups	Case study	Blockchain technology can solve the paradox of platform ecosystem openness by standardizing and automating interactions between multiple participants.
Cennamo and Santaló (2019) [71]	PE is a complex ecosystem of businesses where individual and collective interests are intertwined.	Electronic game platform system	Quantitative analysis	Coordinators play a pivotal role in shaping the success of the PE and exerting influence on its competitive position in the marketplace.
Tavalaei and Cennamo (2021) [72]	PE is a new structure of economic relationships between firms formed through the provision of resources by platform leaders to create value for complementarities.	Mobile application ecosystem	Quantitative analysis	PE members must ensure that they have a unique position in relation to their competitors.
Inoue (2021) [73]	PE is a system or architecture consisting of collections of complementary assets.	Incremental and radical innovation	Quantitative analysis	PEs can benefit from unlimited innovation through a variety of complementary and potentially unlimited pools of external resources.
Cenamor and Frishammar (2021) [70]	PE consists of incompletely designed product and technology “platforms” and complements.	Complementary products	Empirical analysis	The delineation of firm boundaries within PEs is shaped not only by the allocation of tasks among ecosystem players but is also impacted by the innovation strategies pursued by the firms involved.
Murthy and Madhok (2021) [19]	PE is a digital collaboration platform for sponsors and autonomous complementary parties to create value together.	Platform sponsor	FQCA (fuzzy qualitative comparative analysis)	The structure of the platform ecosystem, how participants interact, how it functions, and the opportunities it offers are influenced by the activities of the platform’s sponsors.
Kretschmer et al. (2022) [69]	PE is derived from the interdependence between platforms and complementary component sets.	Platform competition	Literature analysis	The key to competition in the platform ecosystem is to coordinate and manage the various players on the platform.
Thomas et al. (2014) [18]	In a PE, control over the entire product system is released, allowing the integration of different products. The ecosystem introduces market dynamics such as network effects and coordinated dominance through interactions between buyers and sellers.	Concept of architectural leverage and platform	Literature review	The PE is a versatile architectural approach that combines production, innovation, and transactional elements in a many-to-many structure. It harnesses the logic of open systems to create and distribute value through production, innovation, and transactions.

**Table 6 biomimetics-09-00216-t006:** Research theme of selected articles.

Field	Article	Research Theme
Business Ecosystem	[118]	Elucidating concepts and methods for improving BE resilience and antifragility.
	[130]	Elucidating how meta-organizations coordinate user engagement and use advanced AI technology to help businesses create social and economic value.
	[132]	Studying the relationship between a strong platform, cooperative strategies, and building a BE in “Internet+ WEEE collection” means looking at how a powerful platform, teamwork, and the overall system work together to make electronic waste collection more effective in the internet context.
	[127]	Investigating how the structure of BEs influences a company’s performance after facing setbacks.
	[114]	Describing how companies collaborate on open innovation strategies to speed up the technological advancement of their partners, removing obstacles in the BE.
	[129]	Investigating how retailers incorporate digital technologies into their business models to generate value through connections with external partners.
	[126]	Studying how open innovation acts as a mediator and how digitalization capabilities serve as a moderator in the connection between coopetition strategy and sustainable performance within the context of the Belt and Road Initiative.
	[128]	Clarifying the boundary conditions of digital transformation from an organizational change perspective.
	[131]	Exploring entrepreneurial growth in the digital business ecosystem requires examining how enterprises can reconfigure their knowledge base to achieve this objective within the dynamic digital landscape.
Innovation Ecosystem	[108]	Showing how companies cultivate dynamic capabilities to effectively coordinate and manage their IEs.
	[109]	Investigating the evolving requirements of social innovators and the interplay between social IEs and traditional entrepreneurial ecosystems.
	[103]	Examining the formation, evolution, and role of digital IEs in facilitating disruptive innovation through processes such as exaptation.
	[110]	Investigating the critical factors contributing to the stability of green IEs and evaluating the influence of policies on collaborative innovation among multiple agents within the green IEs.
	[107]	Investigating how government support influences the growth of China’s digital economy and the significant role played by the resilience of regional IEs in this development.
	[105]	Examining the consolidation and evolution of Industry 4.0-oriented IEs and how value co-creation occurs within these ecosystems to deliver solutions for the market.
	[112]	Investigating the evolution of the role of multinational corporations in attaining sustainable development goals, particularly as the IE and corporate responsible research and innovation mature over time.
	[106]	Applying the principles of IE and responsible research and innovation to ethical discussions in artificial intelligence (AI) to explore ways of establishing a responsible AI innovation ecosystem.
	[104]	Examining the potential configurations and characteristics of IEs for small and medium-sized enterprises (SMEs) that enable the development and delivery of industrial Smart Products.
	[111]	Exploring the innovation of sustainable and smart products from the standpoint of an IE, considering the collaborative relationships and dynamics among various entities involved in the development and delivery of such products.
	[113]	Examining how equipment suppliers develop suitable ecosystem strategies to achieve digitally enabled process innovation in different industrial customer contexts.
	[96]	Investigating the impact of digital technologies and knowledge digitalization on the broader landscape of technology entrepreneurship and the processes involved in creating new ventures.
Platform Ecosystem	[82]	Examining the strategies through which industrial manufacturers can enhance the value of their platforms by evolving and advancing industrial digital platforms.
	[92]	Investigating the ongoing integration of digital capabilities and how they transform the scale and scope of product platform functions, resulting in a shift in the organizational logic of the firm.
	[89]	Investigating the strategies and mechanisms for developing resilience in digital PEs.
	[87]	Analyzing how a firm in a peripheral role in a PE can redefine its position through a dynamic mix of tangible, representative, and structural initiatives to create and establish an alternative platform.
	[86]	Elaborating on the mechanisms and reasons behind the enduring quality of complementary products within digital PEs.
	[90]	Investigating how users’ preferences for the size and innovativeness of an ecosystem impact the co-creation of value and the strategic dynamics within the PE.
	[91]	Examining the capabilities that companies require to facilitate and balance the processes of value co-creation and value capture within emerging digital PEs.

## Data Availability

Data are contained within the article.

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
