# Peer review of "Comparing Business, Innovation, and Platform Ecosystems: A Systematic Review of the Literature"

_biomimetics, 2024, doi:10.3390/biomimetics9040216_

Round 1

Reviewer 1 Report (Previous Reviewer 1)

Comments and Suggestions for Authors

Thanks for the revision of the submitted paper.
Most of the comments from the last version were revised; however, you need to consider the reliability of the data sets for the Mixed methodology with Citation analysis and Content analysis you conduct.
Please find detailed comments below.  

1. Figure 1 Modification and Explanation Needed (236)

 : In 82 papers for content analysis, 32 papers were utilized, and 27 papers that remained for future reference were categorized. However, among a total of 59 papers, 23 papers were excluded and needed to be explained. Additionally, there are typographical errors within the images that require correction.

 2. Differentiation of BE/IE/PE through Content Analysis

 : Although 32 papers were analyzed for content analysis, only 15 were cited in Table 1 (336).

 3. Explanation of CiteSpace’s Algorithm in detailed(293, 301)

 : Detailed explanation of CiteSpace’s algorithm and assessment of its reliability for this study’s cluster analysis. Furthermore, additional citations from the previously utilized papers are necessary.

 4. Criteria and Rationale for Selecting the Time Period (379)

 : Consider excluding the year 2023 to analyze by year.

Comments on the Quality of English Language

Nothnig to comment in this time. 

Author Response

Thank you for the comments of the academic editor and reviewers. The authors revised the research article “Comparing Business, Innovation, and Platform Ecosystems: A systematic review of the literature.” Based on the reviewer's feedback, we have re-edited the entire manuscript to improve the overall quality of scientific content and grammar. 

The modification description in detail has been included as an appendix to this cover letter. We sincerely acknowledge the three anonymous reviewers for their affirmative and professional comments on this manuscript.

Best regards.

All the authors

Reviewer 2 Report (Previous Reviewer 2)

Comments and Suggestions for Authors

In my opinion, the revised version can be published.

Comments on the Quality of English Language

I have no additional comments 

Author Response

Thank you very much for reviewing our paper and providing your affirmation and support for our work. Your feedback is truly encouraging and will motivate us to continue striving for excellence and enhancing our research capabilities. We are deeply grateful for your recognition and support, which are invaluable to our efforts. Thank you once again for your time and valuable input.

Reviewer 3 Report (Previous Reviewer 3)

Comments and Suggestions for Authors

The paper is too long and difficult to read. Therefore we suggest a shortened version of the manuscript. 

Author Response

Thank you for the comments of the academic editor and reviewers. The authors revised the research article “Comparing Business, Innovation, and Platform Ecosystems: A systematic review of the literature.” Based on the reviewer's feedback, we have re-edited the entire manuscript to improve the overall quality of scientific content and grammar.  

The modification description in detail has been included as an appendix to this cover letter. We sincerely acknowledge the three anonymous reviewers for their affirmative and professional comments on this manuscript.

Best regards.

All the authors

Round 2

Reviewer 3 Report (Previous Reviewer 3)

Comments and Suggestions for Authors

The paper is still very long.

The paper still contains some language errors:

on page 10, Figure 2 Annual publications of the popular 5 Journal.

Comments on the Quality of English Language

The paper still contains some language errors:

on page 10, Figure 2 Annual publications of the popular 5 Journal.

Author Response

Dear Reviewer,

Thank you for your valuable feedback on our paper. We appreciate your time and effort in reviewing our work, and we have carefully considered your comments.

Firstly, we would like to address the issue of the paper's length. We understand that the paper may have appeared overly long, and we have taken steps to address this concern. In the revised version, we have deleted several paragraphs in Section 1 and Section 2.2 that are deemed redundant. We believe that these changes have significantly improved the flow and readability of the paper, while still maintaining the integrity of our research.

Secondly, we have carefully reviewed the language of the paper and corrected any errors that were identified. We have made sure to improve the clarity and accuracy of our writing, ensuring that the revised version is free from grammatical and syntactical mistakes.

We hope that these changes have addressed your concerns and that the revised paper meets your expectations. We are committed to providing high-quality research and appreciate your guidance in helping us improve our work.

Thank you again for your feedback, and we look forward to your continued support.

Best regards,

All the authors

This manuscript is a resubmission of an earlier submission. The following is a list of the peer review reports and author responses from that submission.

Round 1

Reviewer 1 Report

Comments and Suggestions for Authors

The following comments can help improve your manuscript.

1. The manuscript's content seems not to fit the aim and scope of the journal, Biomimetics.  It also did not significantly contribute to understanding or applications of management-related ecosystems.

2. The authors need to check the reference format of Biomimetics. For references in the text, the reference numbers shall be in brackets [ ]. (See MDPI reference guideline. https://mdpires.com/data/mdpi_references_guide_v5.pdf) If you wish to cite a specific paper in the text, the reference shall be formatted as follows: Complementors, as emphasized by Pierce [9], ~. I found some incorrectly formatted references. I recommend the authors thoroughly reexamine the text's references before resubmitting the revised manuscript.

3. The authors summarize previous research in lines 191 to 193 without citing references. Please quote related articles.

4. In lines 271 to 273, the authors described that the search period was set between 1993 to July 16, 2023, without explaining the proper reasons behind such a period. I am guessing that the authors may have started to extract initial data after July 16, 2023, which might be why the authors set the end of the search period as July 16, 2023. In addition, the authors should have provided proper reasons for why only journal articles published before 1993 were excluded from the analysis.

5. In section 3, the authors illustrate the overall research procedure. Compared with 3.1, in which the authors described how they gathered raw data from WoS, 3.2 and 3.3 failed to provide details of methods used in the manuscript. In addition, I recommend the authors describe the following things. e.g., which tool was used for content analysis? How did the authors conduct qualitative coding? How did the authors verify the qualitative coding result?

6. The authors highlighted phases of the systematic review in Figure 1. The authors used bibliometric and content analyses to overview ecosystem-related literature comprehensively. As shown in Fig. 1, The authors conducted bibliometric analysis on 956 articles; however, content analysis was performed on a minimal number of articles, only 32. Neither section 3 nor section 4 provided the rationale behind why the authors screened twice to finalize target articles for content analysis.

7. Unlike the title of 4.1.2., Co-cited journal network analysis, the authors only focused on highlighting top-cited journals and the number of citations they received using a network diagram and a graph. I suggest the authors modify 4.1.2. to further explain two things: What does the journal co-citation mean? What can we analogize from the analysis result?

 8. In 4.1.3., the authors described a setting for keyword cluster analysis, such as three threshold values. However, a detailed explanation must be included of how the keyword clustering result was derived. For instance, what the threshold values indicate and the clustering algorithms used in Citespace need to be elaborated.

9. In 4.1.4., the keyword co-occurrence analysis result was provided based on three stages, which the authors identified.

10. In Table 1, each cluster has several keywords. For example, #0 Innovation ecosystem includes keywords such as performance, management, etc. However, it is hard to distinguish which references are included or not. Therefore, I recommend the Authors remedy table 1 to be easier to figure out.

Comments on the Quality of English Language

I recommend using a professional English service, such as MDPI's.

Reviewer 2 Report

Comments and Suggestions for Authors

This is a useful manuscript. The authors did much work compiling the relevant literature. I have some critical comments that I'd like to be addressed in the revised version.  

It should be stated somewhere that the manuscript is a personal view of the authors on the problems and not the "final truth".

In section 6. Conclusions I have not found a single conclusion.  Where are the authors' results of their analysis? The shorter specific conclusions the better. The authors only related a bunch of references. Their Figures allow them to come to the conclusions. I recommend the authors to see how conclusions are written in Chemical Reviews or any other high-impact review Journal.

Figure 6 - impossible to read comments.

Lines 363-364 require explanations.

I recommend adding a list of abbreviations at the end of the manuscript.

It is not standard practice to present the initials of the cited authors in the text.

Comments on the Quality of English Language

Some sentences are just word after word without much sense. 

Reviewer 3 Report

Comments and Suggestions for Authors

This study addresses the existing research gap by employing a hybrid methodology that combines bibliometric and content analyses. The systematic review spans the literature from 1993 to 2023, with a primary focus on theoretical studies concerning various ecosystem types, particularly those related to business, innovation, and platform ecosystems. The methodology encompasses a thorough content review of identified literature, utilizing quantitative bibliometric analysis to identify patterns and content analysis for a comprehensive exploration. The key findings of this study involve refining and summarizing the definitions of business, innovation, and platform ecosystems, highlighting both commonalities and distinctions. Notably, the research uncovers shared characteristics such as openness and diversity across these ecosystems, while emphasizing significant differences in terms of participants and objectives. Additionally, the paper explores the interconnections within these three ecosystem types, providing insights into their dynamics and paving the way for discussions on future research directions.

The paper is well-written and the contribution is original. Therefore we propose accepting the paper in its present form.